# *In vivo* multiscale analyses of spring viremia of carp virus (SVCV) infection: From model organism to target species

**Sandra Souto**[1], **Raquel Lama**[1,2], **Emilie Mérour**[2], **Manon Mehraz**[3], **Julie Bernard**[2], **Annie Lamoureux**[2], **Sarah Massaad**[3], **Maxence Frétaud**[2], **Dimitri Rigaudeau**[3], **Jean K. Millet**[2], **Christelle Langevin**[3]*, **Stéphane Biacchesi**[2]*

1 Microbiology and Parasitology, Universidade de Santiago de Compostela, Santiago de Compostela, Spain,
2 Université Paris-Saclay, INRAE, UVSQ, Virologie et Immunologie Moléculaires, Jouy-en-Josas, France,
3 Université Paris-Saclay, INRAE, Infectiologie Expérimentale des Rongeurs et des Poissons, Jouy-en-Josas, France

☙ These authors contributed equally to this work.
* christelle.langevin@inrae.fr (CL); stephane.biacchesi@inrae.fr (SB)

**Data Availability Statement:** All relevant data are within the manuscript and its Supporting Information files.

## Abstract

Spring viremia of carp virus (SVCV) has a broad fish host spectrum and is responsible for a disease that generally affects juvenile fishes with a mortality rate of up to 90%. In the absence of treatments or vaccines against SVCV, the search for prophylactic or therapeutic solutions is thus relevant, particularly to identify solutions compatible with mass vaccination. In addition to being a threat to aquaculture and ecosystems, SVCV is a unique pathogen to study virus-host interactions in the zebrafish model. Establishing the first reverse genetics system for SVCV and the design of recombinant SVCV (rSVCV) expressing fluorescent or bioluminescent proteins adds a new dimension for the study of these interactions using innovative imaging techniques. The infection by bath immersion of zebrafish larvae with rSVCV expressing mCherry allows us to define the first SVCV replication sites and the host innate immune responses using different transgenic lines of zebrafish. The fins were found as the main initial sites of infection in both zebrafish and carp, its natural host. Hence, new insights into the physiopathology of SVCV infection have been described. We report that neutrophils are recruited at the sites of infection and persist up to the death of the animal leading to an uncontrolled inflammation correlated with the expression of the pro-inflammatory cytokine IL1β. Tissue damage was observed at the site of initial replication, a likely consequence of virus-induced injury or the pro-inflammatory response. Interestingly, SVCV infection by bath immersion triggers a persistent pro-inflammatory response rather than activation of the antiviral IFN signaling pathway as observed following intravenous injection, highlighting the importance of the route of infection on the progression of pathogenicity. Thus, this model of zebrafish larvae infection by rSVCV offers new perspectives to study in detail virus-host interactions and to discover new prophylactic or therapeutic solutions.

**Funding:** IVIS Spectrum and COPAS were financed by the région Ile-de-France (DIM-1Health). This work is supported by a public grant overseen by the French National Research Agency (ANR) as part of the "Investissements d'Avenir" program (Labex NanoSaclay, reference: ANR-10-LABX-0035) to CL. SS was funded with a postdoctoral grant from Consellería de Cultura, Educación y Universidad, Xunta de Galicia (postdoctoral grant ED481B-2018/002). RL was funded by the Ministerio de Universidades (Spanish Government), and Next Generation EU Programme for the Requalification, International Mobility and Attraction of Talent in the Spanish University System, modality Margarita Salas (postdoctoral grant UP 2021-042). The funders had no role in study design, data collection and analysis, decision to publish, or preparation of the manuscript.

**Competing interests:** The authors have declared that no competing interests exist.

## Author summary

In addition to being a threat to many valuable aquaculture species, spring viremia of carp virus (SVCV) is a model pathogen used to study host immune responses for the zebrafish community. Here, we bring another dimension to these studies thanks to the generation of traceable recombinant viruses by reverse genetics enabling for the first time *in vivo* imaging of SVCV infection. Upon zebrafish inoculation with a recombinant SVCV expressing mCherry, we established a robust and highly reproducible model of infection based on the natural route of infection. This powerful model allows us to define the first SVCV replication sites, viral spreading, target organs as well as host responses using different transgenic lines of zebrafish. We found that the fins were the main initial sites of infection and not the gills as commonly reported. This was confirmed in carp, the natural host of SVCV. New insights into the physiopathology of SVCV infection have been described. Thus, the development of this standardized model of viral infection in zebrafish larvae opens new perspectives to better understand SVCV infection and to discover anti-viral drugs and/or immunomodulating compounds assessed by *in vivo* high content screening while ensuring 3R compliance on animal experimentation.

## Introduction

Aquaculture is the fastest-growing sector in livestock production in the world and is estimated to have reached 87.5 million tons of aquatic animals, encompassing around 425 farmed species [1,2]. Diverse risks and constraints represent increasing challenges for the sustainability of the aquaculture industry which include the growing production of nutrients, chemical pollution, climate change, lower food abundance, and the emergence of disease outbreaks in farming systems [3,4]. Among the solutions, vaccines have already proven their high potential to improve animal health and welfare while reducing the reliance on antimicrobial use to prevent disease outbreaks. However, vaccines against aquatic viral pathogens are either non-existent or offer only limited protection in the field and might lead to increased virulence of targeted viruses in the future [5,6]. Several detrimental viral diseases in farmed fish populations such as rhabdovirus infections frequently cause substantial mortality, reduced welfare, and large economic losses representing a limiting factor for the sustainability of aquaculture. Moreover, they threaten the stocks of wild fish with important environmental consequences once accidentally introduced in a naïve environment [7]. In addition, climate change will likely modify fish feeding habits and the availability of natural food resources leading to higher sensitivity to disease outbreaks [8].

*Sprivivirus cyprinus*, commonly known as spring viremia of carp virus (SVCV), belongs to the *Sprivivirus* genus in the *Rhabdoviridae* family [9]. SVCV is listed as notifiable to the World Organization for Animal Health because of its highly infectious nature. It has a broad fish host spectrum (*e.g.* carp, pike, sturgeon, goldfish) and causes a contagious acute hemorrhagic viremia which generally affects juvenile fishes (<1 year of age), with a reported mortality rate of up to 90% during the spring when water temperature ranges between 11 and 17˚C. Efficient SVCV replication has been achieved in a wide range of cell types derived from fish, birds, and mammals at temperatures ranging from 4˚C to 31˚C [9] and the virus has even been isolated from diseased shrimp in Hawaii [10] and dead salamanders imported from China to the United States [11]. Viral transmission is horizontal and occurs through contaminated water by direct excretion of the virus from infected fish. Following experimental bath infection of carp, the virus was initially detected in the gills suggesting they were the first targeted organs [9].

The virus then enters the vascular network and spreads to the liver, kidney, spleen, and digestive tract, leading to the death of infected fish. The presence of the virus is reported in ponds throughout the Northern Hemisphere in the main cyprinid-producing countries (Asia, Eastern Europe, and North America). Common carp farming ranks fourth in finfish inland aquaculture worldwide with production reaching 4.2 million tons in 2020 [1]. The production systems of carp are semi-intensive or intensive fish farms, intended either for food consumption or for the production of ornamental varieties, such as koi carp of important value. In the absence of treatments or vaccines against SVCV, the search for prophylactic or therapeutic solutions, such as live attenuated vaccines or natural molecules with immunomodulatory effects on fish immunity, is thus relevant, particularly to identify solutions compatible with mass delivery needs (*i.e.* bath immersion rather than injection of vaccines). In addition, SVCV is used as a model virus to study host-pathogen interactions in zebrafish (*Danio rerio*). Indeed, zebrafish can be experimentally infected with SVCV and is widely used as a biomedical model because of its small size and optical transparency. These characteristics are particularly appreciated for *in vivo* screening of therapeutic molecules. Moreover, the innate immune system, the first line of defense against viral invasion, is remarkably well conserved between zebrafish and humans making zebrafish a powerful model for antiviral research [12–14].

The genome of SVCV consists of a single molecule of negative-sense, single-stranded RNA of approximately 11 kb [15]. The genomic RNA encodes five genes in the order 3′-N-P-M-G-L-5′ encoding a nucleoprotein (N), a phosphoprotein (P), a matrix protein (M), a glycoprotein (G) and an RNA-dependent RNA polymerase (L), respectively. The N protein tightly encapsidates the genomic and antigenomic viral RNAs to form together with the viral polymerase L and its cofactor, the P protein, a ribonucleoprotein (RNP) complex that ensures viral gene expression and genome replication. In the virion, the helical RNP complex is condensed by the M protein which links it to the viral envelope derived from the host cell, in which the unique G glycoprotein is inserted. However, although the genomic structure of SVCV has been characterized and several reverse genetics systems have been described for many viruses belonging to the *Rhabdoviridae* family [16], there are still no such systems available allowing genetic manipulation and the production of recombinant SVCV particles.

In the present study, we describe the first reverse genetics system for SVCV and the generation of highly traceable viruses allowing real-time visualization of infection processes in the target host (carp) and the innate immune responses in a model fish (zebrafish). These robust and reproducible infection models open the possibility to uncover the initial sites of viral replication, representing privileged locations of entry into the host organism, and to characterize new insights into the pathophysiology of SVCV infection. The development of this real-time traceable model of zebrafish infection by SVCV provides a powerful pipeline for antiviral and immunomodulatory drug screening.

## Results

### SVCV reverse genetics system and recovery of tracer viruses

A strain widely considered as a prototype and named Fijan (NCBI accession no. AJ318079.1) was used as the basis for developing our reverse genetics system for SVCV (*Sprivivirus cyprinus*). To contextualize the phylogenetic position of this prototype strain with other available genomic sequences of SVCV, a whole-genome analysis was performed on 16 SVCV strains along with the genomic sequence of a more distantly related virus, Pike fry rhabdovirus (*Sprivivirus esox*) (Fig 1A). The analysis reveals that the Fijan strain which infects carp and originated from the former Yugoslavia in 1971 forms a basal clade to the other SVCV genomes analyzed. Interestingly, the more recently characterized SVCV strains are able to infect an

## A

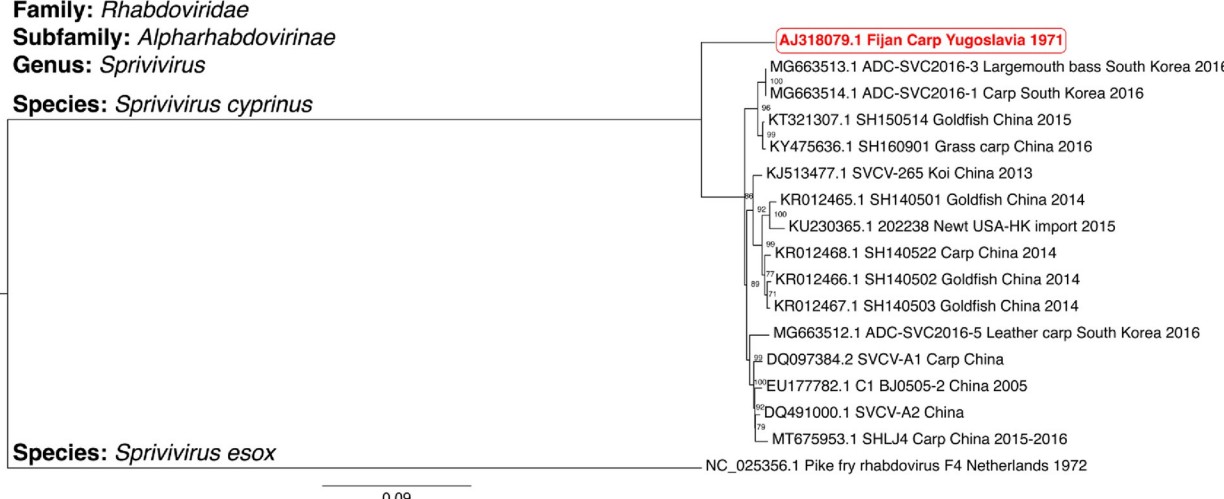

**Whole-genome analysis**

**Family:** *Rhabdoviridae*
**Subfamily:** *Alpharhabdovirinae*
**Genus:** *Sprivivirus*
**Species:** *Sprivivirus cyprinus*

AJ318079.1 Fijan Carp Yugoslavia 1971
MG663513.1 ADC-SVC2016-3 Largemouth bass South Korea 2016
MG663514.1 ADC-SVC2016-1 Carp South Korea 2016
KT321307.1 SH150514 Goldfish China 2015
KY475636.1 SH160901 Grass carp China 2016
KJ513477.1 SVCV-265 Koi China 2013
KR012465.1 SH140501 Goldfish China 2014
KU230365.1 202238 Newt USA-HK import 2015
KR012468.1 SH140522 Carp China 2014
KR012466.1 SH140502 Goldfish China 2014
KR012467.1 SH140503 Goldfish China 2014
MG663512.1 ADC-SVC2016-5 Leather carp South Korea 2016
DQ097384.2 SVCV-A1 Carp China
EU177782.1 C1 BJ0505-2 China 2005
DQ491000.1 SVCV-A2 China
MT675953.1 SHLJ4 Carp China 2015-2016

**Species:** *Sprivivirus esox*

NC_025356.1 Pike fry rhabdovirus F4 Netherlands 1972

0.09

## B

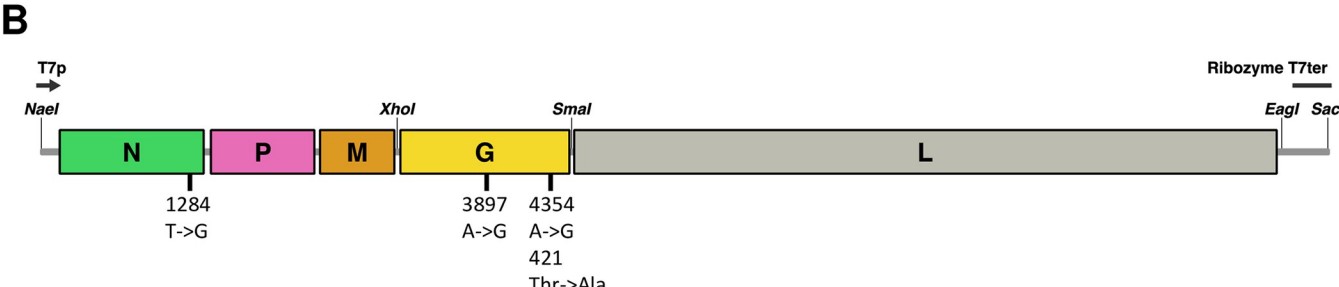

**Fig 1. SVCV genome and infectious cDNA construct. A. Whole-genome SVCV phylogenetic analysis**. The complete genomic sequences of 16 SVCV isolates (*Sprivivirus cyprinus*) and Pike Fry Rhabdovirus (*Sprivivirus esox*) were obtained from NCBI Genbank. The SVCV sequence highlighted in red corresponds to the Fijan strain (Genbank #AJ318079.1) and was used to establish the SVCV reverse genetics system. Accession numbers are provided in each branch of the phylogenetic tree and in Table 1. The genomes were aligned using MUSCLE and the choice of substitution model was performed using MEGA X. A Maximum-Likelihood phylogenetic tree was inferred from the alignment using PhyML with the GTR+G substitution model and 1000 bootstrap replicates. Numbers at nodes indicate percent bootstrap support. The tree was drawn to scale using FigTree 1.4.4 and rooted using Pike fry rhabdovirus. Branch lengths correspond to the number of substitutions per site. The host species, date and country of origin are shown when found in the sequence metadata. B. Construction of the pSVCV plasmid encoding the complete antigenomic RNA of the SVCV Fijan strain. Five overlapping cDNA fragments (numbered 1 to 5) covering the complete Fijan strain antigenome were generated by RT-PCR with primers described in Table 2 and were assembled in pBluescript SK– following appropriate restriction enzyme digestion. In the final construct, named pSVCV, the leader end of the antigenome cDNA is flanked by the T7 promoter (T7), and the trailer end is fused to the hepatitis delta virus ribozyme followed by a terminator for T7 RNA polymerase (T7t). The complete antigenomic cDNA was designed to create *XhoI*, *SmaI* and *EagI* unique restriction sites in the M-G, G-L intergenic and trailer regions, respectively (see S1 Fig). In addition, this genome differs from the genome of the Fijan reference strain by three nucleotide substitutions in N and G genes.

extensive range of hosts, including carp, grass and leather carp, koi, goldfish, largemouth bass, and even a newt (an amphibian) imported from Hong Kong to the United States [11] (Table 1). The basal topology and phylogenetic distance of the Fijan strain compared to the other SVCV strains is likely due to its sampling location (former Yugoslavia) and date (1971) which is several decades earlier than the other available genomic sequences that were sampled mostly in Asia and with dates ranging from 2005 to 2016 (Table 1 and Fig 1A).

The complete RNA genome of the prototypical Fijan strain (Fig 1A) was amplified by reverse transcription and PCR as 5 overlapping fragments using specific primers (Table 2). Each PCR product was verified by Sanger sequencing. The final cDNA genome differs from

**Table 1. Accession numbers of complete genome sequences used in phylogenetic analysis.**

| NCBI accession no. | Isolate name | Host species | Sampling location | Sampling date |
|---|---|---|---|---|
| AJ318079.1 | Fijan | Carp (*Cyprinus carpio*) | Yugoslavia | 1971 |
| EU177782.1 | BJ0505-2 | Pooled fish sample | China | 2005 |
| DQ097384.2 | SVCV-A1 | Carp (*Cyprinus carpio*) | China | n.d. |
| KJ513477.1 | SVCV-265 | Koi carp (*Cyprinus rubrofuscus*) | China | 2013 |
| KR012466.1 | SH140502 | Goldfish (*Carassius auratus*) | China | 2014 |
| KR012465.1 | SH140501 | Goldfish (*Carassius auratus*) | China | 2014 |
| KR012467.1 | SH140503 | Goldfish (*Carassius auratus*) | China | 2014 |
| KR012468.1 | SH140522 | Carp (*Cyprinus carpio*) | China | 2014 |
| KU230365.1 | 202238 | Chinese fire belly newt (*Cynops orientalis*) | USA (imported from Hong Kong, China) | 2015 |
| KT321307.1 | SH150514 | Goldfish (*Carassius auratus*) | China | 2015 |
| KY475636.1 | SH160901 | Grass carp (*Ctenopharyngodon idella*) | China | 2016 |
| MG663513.1 | ADC-SVC2016-3 | Largemouth bass (*Micropterus salmoides*) | South Korea | 2016 |
| MG663514.1 | ADC-SVC2016-1 | Carp (*Cyprinus carpio*) | South Korea | 2016 |
| DQ491000.1 | SVCV-A2 | n.d. | China | n.d. |
| MG663512.1 | ADC-SVC2016-5 | Leather carp (*Cyprinus nudus*) | South Korea | 2016 |
| MT675953.1 | SHLJ4 | Carp (*Cyprinus carpio*) | China | 2015–2016 |
| NC_025356.1 | Isolate F4 | Pike fry (*Esox lucius*) | The Netherlands | 1972 |

n.d.: not determined

**Table 2. Primers for cDNA SVCV constructs.**

| Primer | Sequence (5' to 3')[a] | Location[b] | Restriction site |
|---|---|---|---|
| NAET7SVC | gccggcTAATACGACTCACTATAGGGACGAAGACAAATAAACCATTG | 1–21 | *NaeI* |
| KPNSVC | ggtaccTTCCTCGTTCTTTTTCCCTATGCG | 1986–2015 | *KpnI* |
| KPXHSVC | ggtaccAGCAAGCAGGTCTCCAAACCTAGGG | 2010–2040 | *KpnI* |
| XHOSVC | ctcgagGAACATCAATCTCAATTTAATCTCCC | 3038–3069 | *XhoI* |
| XHSMSVC | ctcgagATATGAAAAAAACTAACAGACATC | 3064–3093 | *XhoI* |
| SMASVC | cccgggCTGTCTCTCAAATAAAGACCGC | 4608–4636 | *SmaI* |
| SMEAGSVC | cccgggACCAAGATATTATCTCAATAGATG | 4631–4660 | *SmaI* |
| EAGSVC | cggccgTTCTATTCTACCCATGTCCCAGATTCTC | 10948–10981 | *EagI* |
| EAGRIBSVC | cggccgTTGTGTAGTATGAAAAAAACTGGATTTGTAGTCTTCGTGGGTCGGCATGGCATCTCCACCTCC | 10976–11019 | *EagI* |
| T7TERMSAC | gagctcCGGATATAGTTCCTCCTTTCAGCAAAAAACCC | | *SacI* |
| Cherry_Fw | cccggg*GTTGTATGAAAAAAACTAACAGAGATC***ATG**GTGAGCAAGGGCGAGGAGG | | *SmaI* |
| Cherry_Rv | cccggg*TTGAGATT***TTA**CTTGTACAGCTCGTCCATGCC | | *SmaI* |
| GFPmaxMG_Fw | cccggg*GTTGTATGAAAAAAACTAACAGAGATC***ATG**CCCGCCATGAAGATCGAG | | *SmaI* |
| GFPmaxMG_Rv | cccggg*TTGAGATT***TCA**TCGAGCTCGAGATCTGGCGAAGG | | *SmaI* |
| ffLUC_Fw | cccggg*GTTGTATGAAAAAAACTAACAGAGATC***ATG**GAAGACGCCAAAAACATAAAGAAAGGC | | *SmaI* |
| ffLUC_Rv | cccggg*TTGAGATT***TTA**CACGGCGATCTTTCCGCCCTTC | | *SmaI* |
| akaLUC_Fw | cccggg*GTTGTATGAAAAAAACTAACAGAGATC***ATG**GAAGATGCCAAAAACATTAAGAAGGG | | *SmaI* |
| akaLUC_Rv | cccggg*TTGAGATT***TTA**CACGGCGATCTTGCCGTCC | | *SmaI* |
| Lsvcv_Fw | GATGATAATACC**ATG**TTTGAATGGGAGAGTCAAGATACTCC | | |
| Lsvcv_Rv | GGTGGTGGTGCTCGA**CTA**TTCTACCCATGTCCCAGATTCTC | | |
| Nsvcv_Fw | GATGATAATACC**ATG**AGTGTCATTCGGATCAAAACAAATGC | | |
| Nsvcv_Rv | GGTGGTGGTGCTCGA**TTA**TCCATAGGTTTGTTTTATCCATTTGCC | | |
| Psvcv_Fw | GATGATAATACC**ATG**TCTCTACACTCAAAATTGTCAGAAAGC | | |
| Psvcv_Rv | GGTGGTGGTGCTCGA**CTA**TAACCTGTATTTTTGATACAACTTATTGTAC | | |

[a] Gs and Ge (gene start/gene end) signals are shown in *italics*. Start/Stop codons are in **bold**. Restriction enzyme sites are in lowercase.
[b] SVCV Fijan reference strain #AJ318079.1.

the genome of the Fijan reference strain (GenBank # AJ318079) by three additional restriction enzyme (RE) sites, a *XhoI* and a *SmaI* RE sites in the non-coding intergenic regions between the M and G genes and the G and L genes, respectively, and an *EagI* RE site in the trailer region (Fig 1B and Materials and methods). In addition, three nucleotide substitutions were detected, two are synonymous mutations and are located in the N and G genes and one is a missense mutation in the G gene from threonine to alanine substitution at position 421 of the glycoprotein. The five fragments were assembled and in the final construct, named pSVCV, the SVCV cDNA antigenome is under the control of the T7 promoter sequence at its 5'-end and fused to a ribozyme sequence derived from the hepatitis δ virus followed by a T7 terminator sequence (S1 Fig) at its 3'-end.

To produce tracer viruses, the pSVCV antigenomic cDNA was modified by the insertion of additional gene(s) encoding fluorescent or bioluminescent proteins. A prerequisite for the expression of a foreign open reading frame (ORF) inserted into a mononegavirus genome is that it should be flanked by appropriate gene start (GS) and gene end (GE) signals for direct transcription by the viral polymerase. Putative consensus sequences of the SVCV GS and GE motifs were determined by aligning the complete genomic sequences of 16 SVCV strains, that were available in GenBank. The intergenic regions within the SVCV genome were compared (S2A Fig) to obtain the minimal consensus sequence DDDRTATGAAAAAAACTAACAGA-SATCATG (with D = G, A or T, R = G or A, and S = G or C) which comprises the untranslated region of SVCV gene containing the transcriptional termination/polyadenylation GE sequence, TATGAAAAAAA, and the transcription initiation GS sequence, AACAGASAT-CATG (with S = G or C). This minimal consensus sequence was then used in all expression cassettes described above for the insertion of an additional gene in SVCV genome either in the intergenic region between M and G genes or between G and L genes using the unique *XhoI* and *SmaI* RE sites, respectively (S2B and S2C Fig). Thereby, five pSVCV constructs were generated containing an expression cassette encoding the mCherry red fluorescent protein or luciferase (LUC) proteins (firefly luciferase ffLUC and Akaluciferase akaLUC) inserted in the SVCV genome between M and G genes or G and L genes, leading to pSVCV-mCherry M/G, pSVCV-mCherry G/L, pSVCV-ffLUC M/G, pSVCV-ffLUC G/L, and pSVCV-akaLUC G/L (Fig 2A). In addition, a pSVCV-GFPmaxCherry construct was designed with two expression cassettes inserted in both positions and encoding a green fluorescent protein (GFPmax) and a mCherry protein (Fig 2A). The total genomic lengths of the encoded recombinant rSVCV-Cherry, rSVCV-GFPmaxCherry and rSVCV-LUC, are 11,711 nt, 12,496 nt and 12,713 nt, respectively, thus 6.3%, 13.4% and 15.4%, respectively, larger than the naturally occurring genome of the SVCV Fijan strain (11,019 nt).

Interestingly, baby hamster kidney 21 (BHK-21) cells were previously shown to be sensitive to SVCV infection [17]. Therefore, rSVCV were generated by transfecting BHK-21-derived BSRT7/5 cells, that constitutively express T7 RNA polymerase [18], with a mixture of four plasmids, pSVCV and the helper plasmids: pSVCV-N, pSVCV-P and pSVCV-L, which encode the nucleoprotein N, the phosphoprotein P, and the RNA-dependent RNA polymerase L, respectively. After transfection, the cells were incubated at 37°C for 6 to 12 h, the transfection mix was then removed and the cells were shifted to 25°C for 7 additional days. Recombinant viruses were finally amplified on fish EPC cells incubated at 25°C. All rSVCV were readily recovered with final titers of $2 \times 10^7$ PFU/mL similar to the wild-type (wt) virus. As shown in Fig 2B–2D, tracer viruses were readily detected in cell culture thanks to fluorescent and bioluminescent proteins expressed during viral replication.

In order to evaluate the genetic stability of such traceable viruses *in vitro*, rSVCV-mCherry was passaged 10 times in EPC cells. After these 10 iterative passages, the reporter gene was still present and contained no mutations and the expression of the mCherry remained intact, demonstrating the stability of such a traceable virus (S3 Fig).

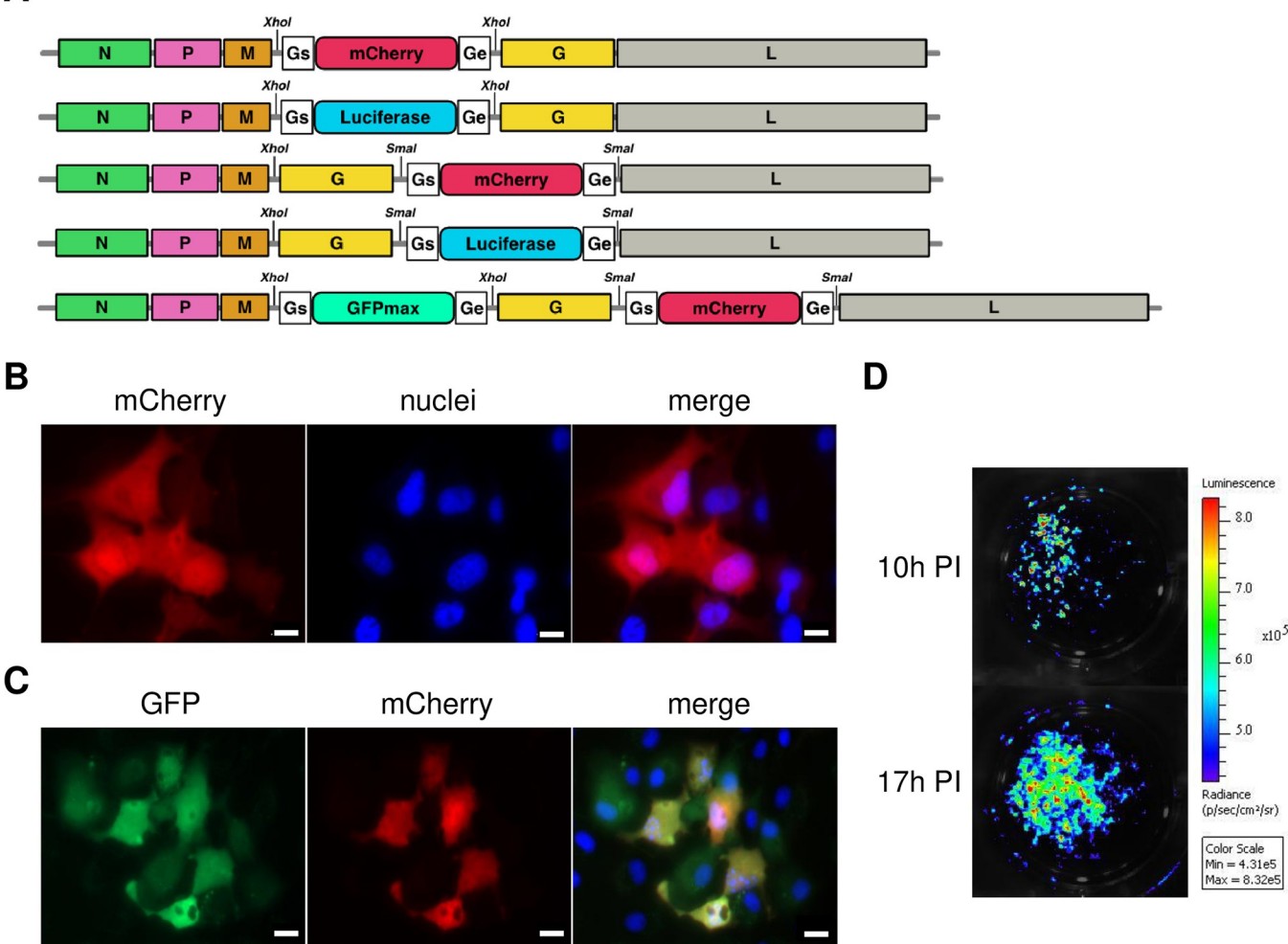

**Fig 2. Insertion of a transcription cassette encoding tracer proteins in the SVCV genome.** A. As described in the Materials and methods, an expression cassette encoding mCherry, firefly luciferase (ffLUC) or akaluciferase (akaLUC) was inserted in the M-G and G-L intergenic regions flanked by additional GS and GE transcription signals of SVCV using the unique *XhoI* and *SmaI* restriction enzyme sites, respectively, leading to the final construct pSVCV-mCherry M/G, pSVCV-ffLUC M/G, pSVCV-mCherry G/L, pSVCV-ffLUC G/L and pSVCV-akaLUC G/L (not to scale). Together with an expression cassette encoding GFPmax between M and G genes, a second expression cassette, as described above, encoding mCherry was inserted in the G-L intergenic region using the unique *SmaI* restriction enzyme site, leading to the final construct pSVCV-GFPmaxCherry that encodes simultaneously both green and red fluorescent proteins. B-C. EPC cells were infected with rSVCV-mCherry (B) or rSVCV-GFPmaxCherry (C) at a final MOI of 0.1. The cells were incubated at 25°C for 24 hours. Live cell monolayers were then visualized with a UV-light microscope after nuclei staining with Hoechst solution in the cell culture medium. 63× objective. Scale bars, 10 μm. D. EPC cells were infected with rSVCV-ffLUC at a MOI of 0.1. At 10 hours post-infection (hpi) and 17 hpi live cells were washed three times with sterile Phosphate Buffer Saline (PBS) and the D-luciferin substrate was added at a concentration of 250 μM. The luminescence was measured using the IVIS Spectrum BL imaging system.

## Virulence of rSVCV in vivo and major portal of entry of SVCV in carp

The virulence of rSVCV was evaluated *in vivo* by bath infection of juvenile carp, and mortalities were recorded daily for 115 days. Control fish were mock infected with cell culture medium under the same conditions. Fig 3 shows the survival curves recorded in each tank. Mortality started as soon as 6 to 10 days post infection and reached 94% to 100% of cumulative mortality after roughly 50 days post infection for rSVCVwt and the Fijan strain. During the acute phase, moribund fish displayed typical petechial hemorrhages of the skin, irrespective of the virus inoculum used. No mortalities were observed in the mock-infected tank. Taken together with the data from *in vitro* infections and titers, these results confirm that rSVCVwt is

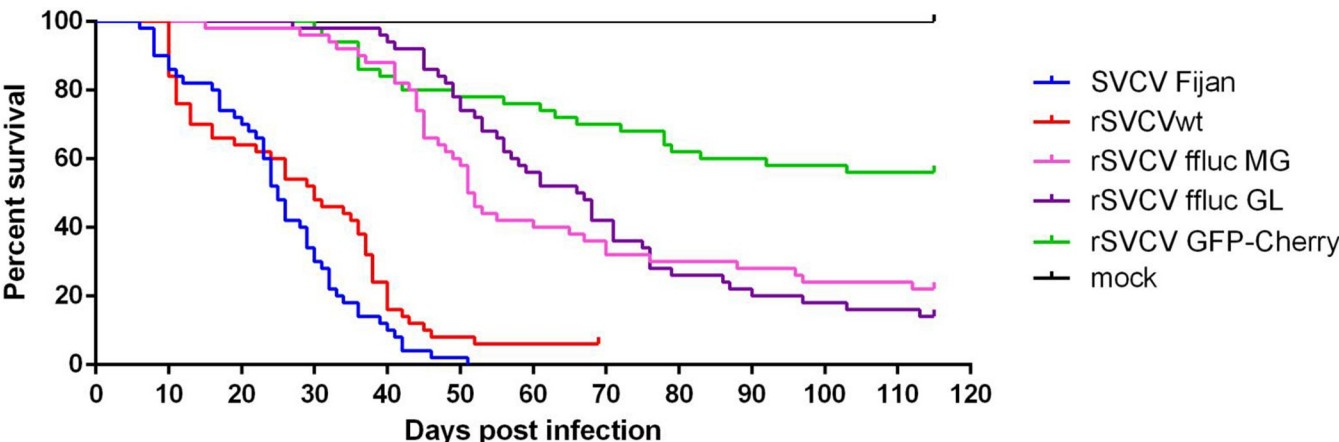

**Fig 3. Experimental carp infection with rSVCVs.** Virulence in carp of rSVCV and rSVCV bearing one or two additional expression cassettes compared to the parental Fijan strain. Juvenile carp [mean weight, 0.81 g (n = 50 per group)] were infected by bath immersion for 2 h in 3 L with $5 \times 10^4$ PFU/mL of each of the indicated rSVCVs and maintained at 10°C. rSVCV is a wild-type recombinant virus derived from the Fijan strain. Mortality was recorded daily and is presented as the percent of survival. Mock, non-infected carp.

fully competent for growth *in vitro* and is as virulent as the SVCV Fijan strain *in vivo*, indicating that the SVCV sequence used in the reverse genetics system is functional and appears to encode a wild-type virus. As expected, recombinant viruses containing one or two additional expression cassettes were attenuated compared to wild-type viruses (SVCV Fijan and rSVCVwt). Mortality induced by rSVCV-GFPmaxCherry, rSVCV-ffLUC MG and rSVCV-ffLUC GL started at 15 to 30 days post infection and reached 44%, 78% and 86% of cumulative mortality, respectively, at day 115 post infection, confirming the attenuating effect of gene insertions and their position along the viral genome as well as the number of the additional expression cassettes as already reported for several members of the *Mononegavirales* [19].

Taking advantage of the recombinant SVCV virus to monitor the infection process *in vivo*, we further investigated the initial replication sites of SVCV in its host by bioluminescence imaging of juvenile carp infected by bath immersion (Fig 4). To improve the detection limits of viral bioluminescent signals recorded in previous studies [20,21], we set up a novel approach based on the ffLUC-derived luciferase (AkaLUC; [22]) with higher signal strength than standard ffLUC and a highly deliverable luciferin analog, Akalumine-HCL, generating near-infrared bioluminescence [23]. Juvenile carp (mean weight, 1.91 g) were infected by bath immersion with rSVCV-akaLUC G/L, expressing the akaLUC, as described in the materials and methods. At 3, 7, and 24 days post infection, 4 fish were randomly transferred to a small tank with water containing Akalumine-HCl substrate. Two hours later, anesthetized fish were subjected to imaging using an IVIS Spectrum BL imaging system. As shown in Fig 4A, luciferase activity was detected in 2 out of 4 fish as soon as 3 days post infection (dpi). On days 3 and 7 post infection, the bioluminescence signals were detected on the skin and the pelvic and caudal fins but none of the fish displayed signals in the gill region. As previously reported for Novirhabdoviruses [21,24] and Koi Herpesvirus [20], the fins and the skin appear to be the initial replication sites of the virus and could therefore constitute the major portal of entry for several fish viruses and not the gills as was often hypothesized [9]. To test the hypothesis of lesion-dependent entry of SVCV in carp, another group of fish had their caudal fins cut prior to infection by immersion with rSVCV-akaLUC G/L. As shown in Fig 4B, 2 fish out of 4 on day 3 post infection and 4 fish out of 4 on day 7 post infection had specific infection foci at the tail cut site reinforcing our hypothesis. Infection with rSVCV mCherry was carried out following the same procedure to identify the SVCV infected cells at cellular resolution by live

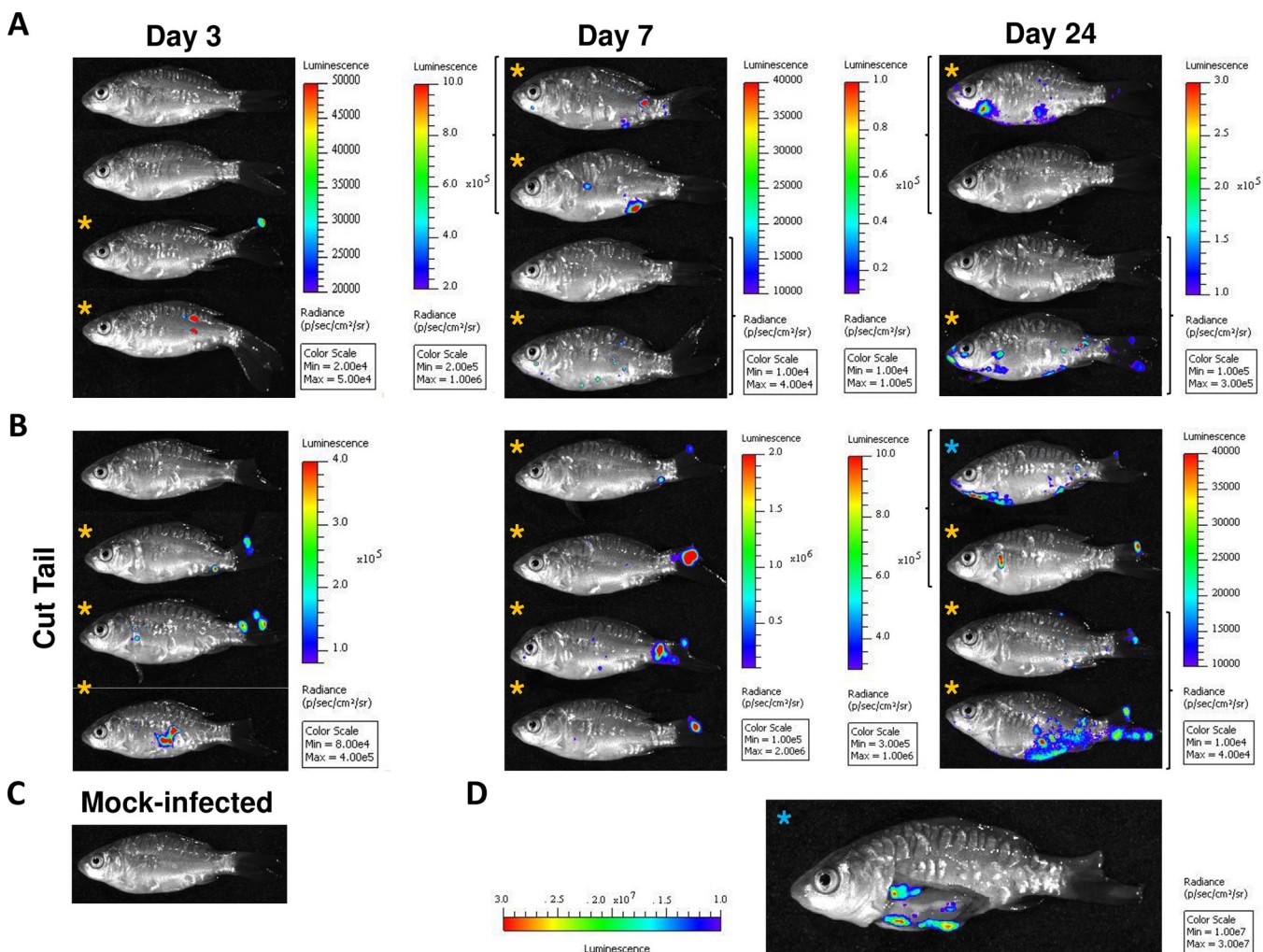

**Fig 4. Kinetics of bioluminescence emission of rSVCV-akaLUC in vivo in carp.** Carp (mean weight, 1.91 g) were divided into two groups with one group infected by immersion with rSVCV-akaLUC G/L (A) as described in the Materials and methods and the other group had their caudal fins cut prior to infection by immersion with rSVCV-akaLUC G/L (B). At 3, 7, and 24 days post infection, fish were randomly harvested and transferred in a small tank with water containing Akalumine-HCl substrate for 2 h at 10°C. Two hours later, anesthetized fish were imaged using an IVIS Spectrum BL imaging system. Mock, non-infected carp (C). Stars indicate carp with detectable bioluminescent foci. Blue stars indicate the same fish before and after dissection.

fluorescence microscopy. mCherry positive cells were visualized at the margin of the caudal fin, between bony rays (interray) at day 3 post infection by stereomicroscopy of bath-infected carp (S4A and S4B Fig). Thus, tissues were fixed, cleared and immunostained to reveal the infected cells (mCherry positive) and cell nuclei (Dapi). 3D confocal images (S1 Video) clearly show the infection of a large number of epithelial cells in the superficial layer of the caudal fin margin. On day 24 post infection, a higher number of bioluminescent foci were detected on infected fish in both groups. In contrast, no luminescence was detected from mock-infected carp used as negative controls at any time point post infection (Fig 4C). At day 24 post infection, some fish were dissected as shown in Fig 4D. Analysis of the dissected fish revealed that bioluminescent signals were detected in previously described target organs, such as hepatopancreas, kidney, spleen, and liver [9].

With the aim of removing any doubt regarding the diffusion of the substrate throughout the body when administered by bath, we compared bioluminescence profiles obtained after

infection with ffLUC and akaLUC virus encoded luciferase reporters. Fish display similar profiles of bioluminescence emission at day 3 post infection when infected with rSVCV-ffLUC and IP-injected with luciferin compared to that infected with rSVCV-akaLUC and bathed in Akalumine-HCl substrate (S5A Fig versus Fig 4). In addition, S5B Fig shows that carp infected with rSVCV-akaLUC but IP-injected with Akalumine-HCl substrate have identical bioluminescence profiles to infected ones bathed in Akalumine-HCl substrate at day 7 and 24 post infection (Fig 4). Examination of bioluminescent signals in internal tissues, exposed upon dissection revealed that the LUC expression profiles were equivalently detected at day 24 post infection irrespective of the route of luciferase substrate administration, indicating proper overall distribution of the substrate for both routes of delivery (S6 Fig).

## A robust and reproducible zebrafish model of rSVCV-mCherry infection

To further investigate the infectious process of SVCV *in vivo*, we developed a zebrafish model to conduct dynamic studies at the level of the whole organism. We combined the advantage of zebrafish larvae transparency and traceable rSVCV to perform live imaging of the infected fish over the course of infection. In contrast to previously reported zebrafish models of SVCV infection established by intravenous injections, we favored a natural route of infection and immersed 3 days post fertilization (dpf) larvae in rSVCV-mCherry viral suspension. Appearance of clinical signs and mortality were assessed at 6, 24, 48, 72, and 96 hours post infection. Larvae mortality was observed as early as 48 hpi reaching 81% at 96 hpi (Fig 5A). To evaluate viral spreading over the same period, infected larvae were anesthetized to detect rSVCV-mCherry signals by fluorescence stereomicroscope imaging. Initial signals were observed at 24 hpi in 68% of larvae displaying red fluorescence signals, which were not detected in the mock-infected fish imaged under the same conditions (Fig 5A). The expression of mCherry fluorescent protein was correlated with the expression of the viral nucleoprotein (Nsvcv) in zebrafish larvae by co-immunolabelling Nsvcv and mCherry. Confocal acquisition of co-immunostained zebrafish larvae showed that tissues expressing mCherry protein were also labeled by the anti-Nsvcv antibody (Fig 5B). The immersion of zebrafish larvae in rSVCV-mCherry inoculum led to active replication of the virus, assessed by titration experiments from individual zebrafish larvae sampled at different times post infection. A 3.9-log increase was observed between 6 and 24 hpi, increasing again by 1.2-log from 24 to 48 hpi, indicating that zebrafish larvae were effectively infected by actively replicating rSVCV (Fig 5C). These results were confirmed by RT-qPCR measurements demonstrating an increase of 4.3-log of the Nsvcv gene expression between 6 and 48 hpi (Fig 5D). All together, these data show that the immersion of zebrafish larvae in rSVCV-mCherry solution allows the establishment of robust and reproducible infections, which are traceable over time by the detection of mCherry fluorescent signals.

## Initial replication sites of rSVCV-mCherry in the zebrafish model

To identify the first sites of rSVCV replication in zebrafish larvae infected by bath immersion, we monitored the appearance of mCherry signal from 6 hpi using *in vivo* fluorescence stereomicroscopy. Initial signals were detected at 24 hpi. Among the 26 larvae analyzed, more than half (n = 16) displayed mCherry fluorescence in the caudal fin (Fig 6A and 6B). The tissues around the mouth and the skin in the trunk region were also infected in 5/26 larvae and 3/26 larvae, respectively (Fig 6A and 6B). Thus, the localization of mCherry fluorescent signals in peripheral tissues from the early stages of infection indicates that the epidermis constitutes the major portal of entry of SVCV in zebrafish larvae, as observed in carp (Fig 4).

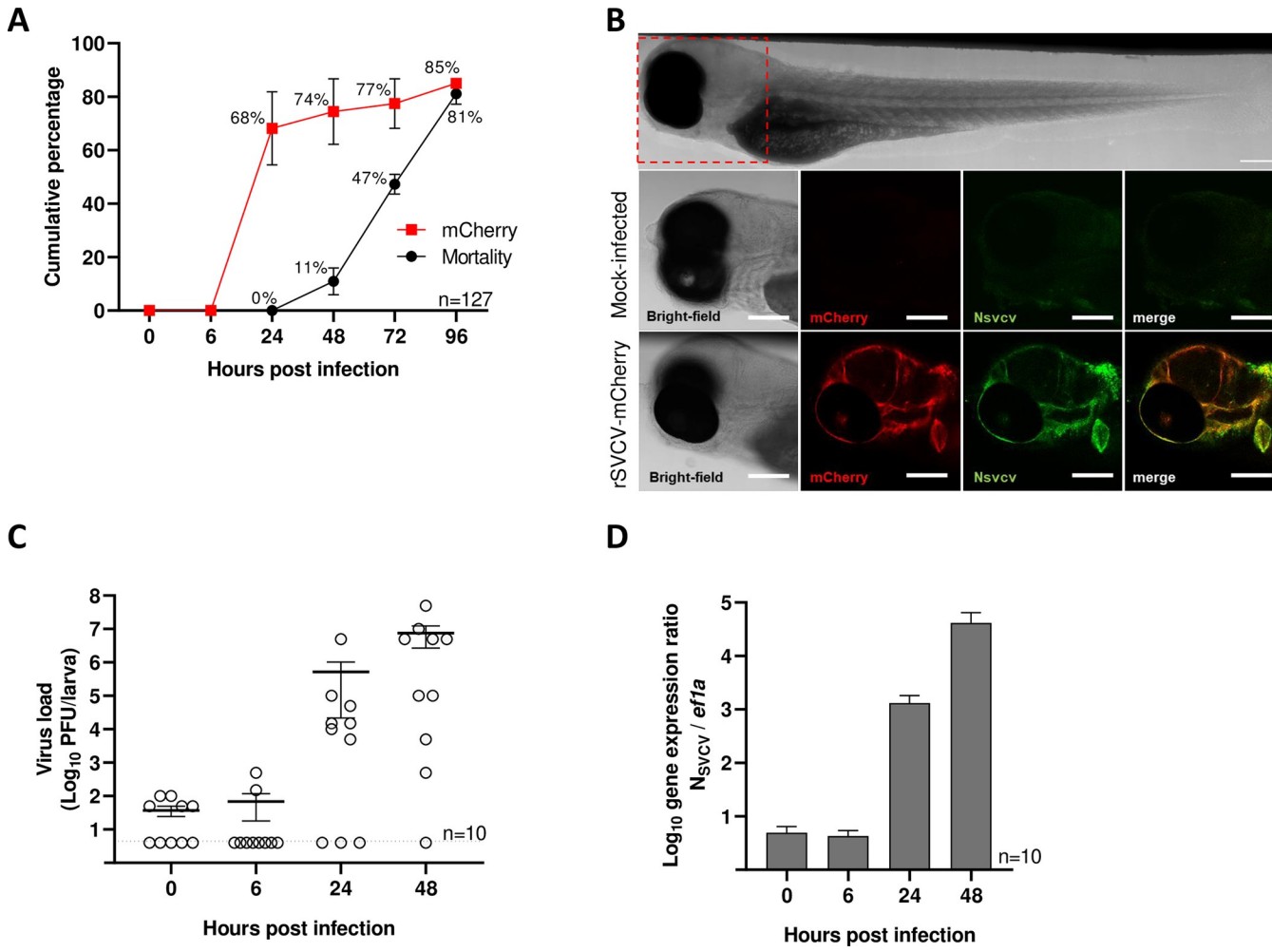

**Fig 5. Zebrafish larvae model of rSVCV-mCherry infection by bath immersion.** A. Nacre zebrafish larvae (a total of n = 147 from three independent experiments) at 3 dpf were infected by bath immersion with rSVCV-mCherry ($2 \times 10^6$ PFU) and incubated at 24°C. Fish were observed at different times post infection for the detection of mCherry fluorescent foci with a fluorescent stereomicroscope and mortalities were recorded daily. Mortality is presented as the mean of cumulative percent of dead larvae recorded in three independent experiments as well as the percent of mCherry positive larvae. No mortalities were recorded in the mock-infected group (a total of n = 30 from three independent experiments). B. Colocalization of mCherry protein and SVCV nucleoprotein (N). Mock infected and SVCV infected larvae were co-immunostained with antibodies raised against mCherry (in red) and SVCV nucleoprotein (in green). Scale bars: 200 μm. C. Virus load in individual larva. At different times post infection, individual larvae (n = 10 in 2 independent infections) were randomly harvested and virus load was determined by plaque titration in EPC cells. Means are shown together with standard errors. D. rSVCV-mCherry replication in zebrafish larvae. Five groups of 5 larvae from 2 independent experiments were randomly harvested at different times post infection and gene expression was analyzed by RT-qPCR. Virus loads are expressed as the ratio of mRNA copy of SVCV nucleoprotein to *ef1a* housekeeping gene.

We next exposed juvenile fish to the same conditions of infections (*i.e.* immersion in rSVCV-mCherry suspension) to determine the impact of zebrafish metamorphosis (change of the body shape, skin structure, appendage morphology, functional gills as respiratory organ [25] and immune maturity) on the sensitivity to the virus. Upon SVCV infection, the death of juvenile fish (25 dpf) occurred between 2 and 8 dpi reaching 73% of mortality at the last infection time point (S7A Fig). Viral load measurements established from individual whole fish, showed an active replication of the virus marked by a 4-log increase between 6 and 24 hpi, and a 3.1-log increase from 24 to 48 hpi (S7B Fig). Observation of infected fish at 24 hpi shows the detection of mCherry positive cells on the superficial tissues of the mandible, the anal and the caudal fins (S7C Fig).

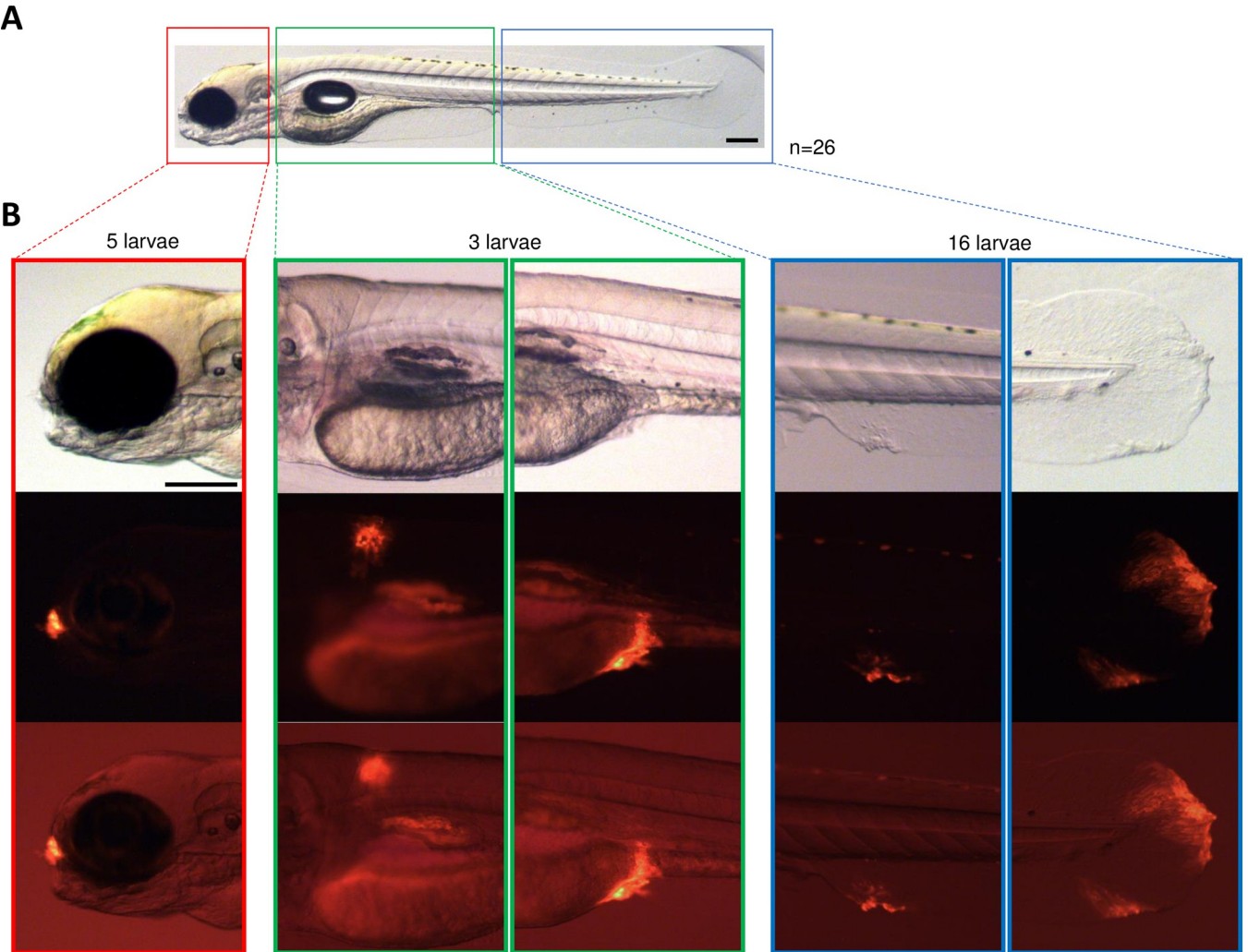

**Fig 6. Detection of mCherry fluorescent foci in larvae at early stages of SVCV infection.** A. Example of a zebrafish larva at 4 dpf in which the anatomical segments corresponding to the head, trunk, and tail are indicated. Scale bars: 200 μm. B. Details of the different foci of infection that can be observed in larvae infected at 24 hpi with $2 \times 10^6$ PFU of rSVCV-mCherry. Of the 26 photographed larvae, 5 larvae had fluorescent foci in the head, 3 larvae in the trunk, and 16 larvae in the tail (2 larvae without detectable fluorescence). Pictures are presented in the following order: brightfield, red fluorescence, and merge of the two previous acquisitions. Magnification 6.3×. Scale bars: 200 μm.

## SVCV viral tropism upon challenge by immersion

From 48 hpi, zebrafish larvae presented an increase of mCherry-positive cells in contact with initial replication sites (visible at the tail fin) associated with a spreading to SVCV main target organs such as the head kidney as shown in Fig 7A. The overall increase in signal intensity is accompanied by a widespread distribution reaching the vascular network, the heart, the intestine, and the liver (Fig 7A). Image analyses establish that at 24 hpi, the majority of fish expressed mCherry fluorescence in the tail before diffusion in the trunk at 48 hpi and in the head at 72 hpi (Fig 7B). At 72 hpi, the mCherry signals significantly increased in all regions and more than 40% of fish were detected as mCherry positive in the head (Fig 7B).

Additional experiments were conducted to evaluate the impact of the route of infection by SVCV on viral tropism. We thus compared the natural route of infection (immersion) to the intravenous (IV) injection in the Duct of Cuvier, already reported in the literature [26]. IV

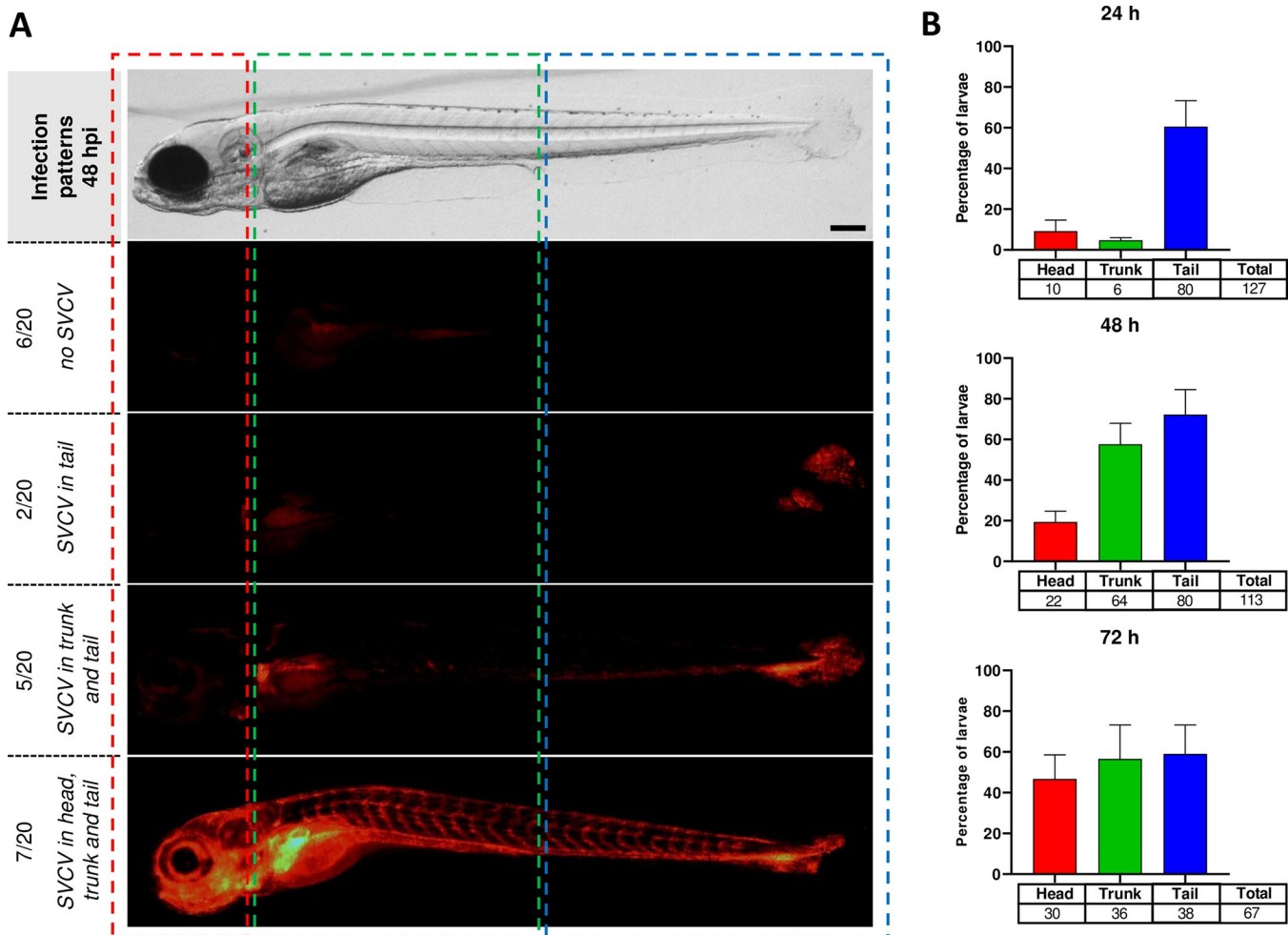

**Fig 7. Viral spread in larvae after bath infection.** A. Whole-larvae infection pattern observed in rSVCV-mCherry infected larvae at 48 hpi. Of the 20 photographed larvae, 2 larvae had only fluorescent foci in the tail, 5 larvae in the trunk and tail, and 7 larvae in the head, trunk, and tail, (6 larvae without detectable fluorescence in the stereomicroscope). Scale bar: 200 μm. B. Tracking of the distribution of the mCherry signal in larvae body (head, trunk, tail) recorded in the same experiment at 24, 48, and 72 hpi (a total of n = 127, data are presented as the mean of 3 independent experiments with standard errors).

injection route shortened the occurrence of the peak of cumulative percent of mortality from 96 hpi to 48 hpi (bath vs IV) and decreased the survival rate leading to the death of 96.6% of the larvae, which was reduced to 81% in the immersion model (Figs 5A and S8A). However, the first mCherry signals did not appear in the injected fish before 24 hpi. In line with these results, we observed higher replication rates as measured by expression of the Nsvcv viral gene by RT-qPCR between 6 and 24 hpi (Figs 5D and S8B). Overall, the fluorescence imaging performed on zebrafish larvae injected with rSVCV-mCherry at 24 hpi revealed a red fluorescence expression pattern almost similar to the one described in zebrafish infected by immersion at later time points (from 48 hpi), with a distribution in the vasculature network, the heart, the kidney, but with the notable exception of the caudal fin fold where the virus was never observed (S8C Fig).

To gain in depth and power of analysis, zebrafish infection was further quantified with complex object parametric analysis and sorting (COPAS) technology. This continuous flow equipment allows the analysis of zebrafish larvae for the detection of their size, optical density, and fluorescent signals while ensuring viability. The COPAS analysis was performed at 6, 24,

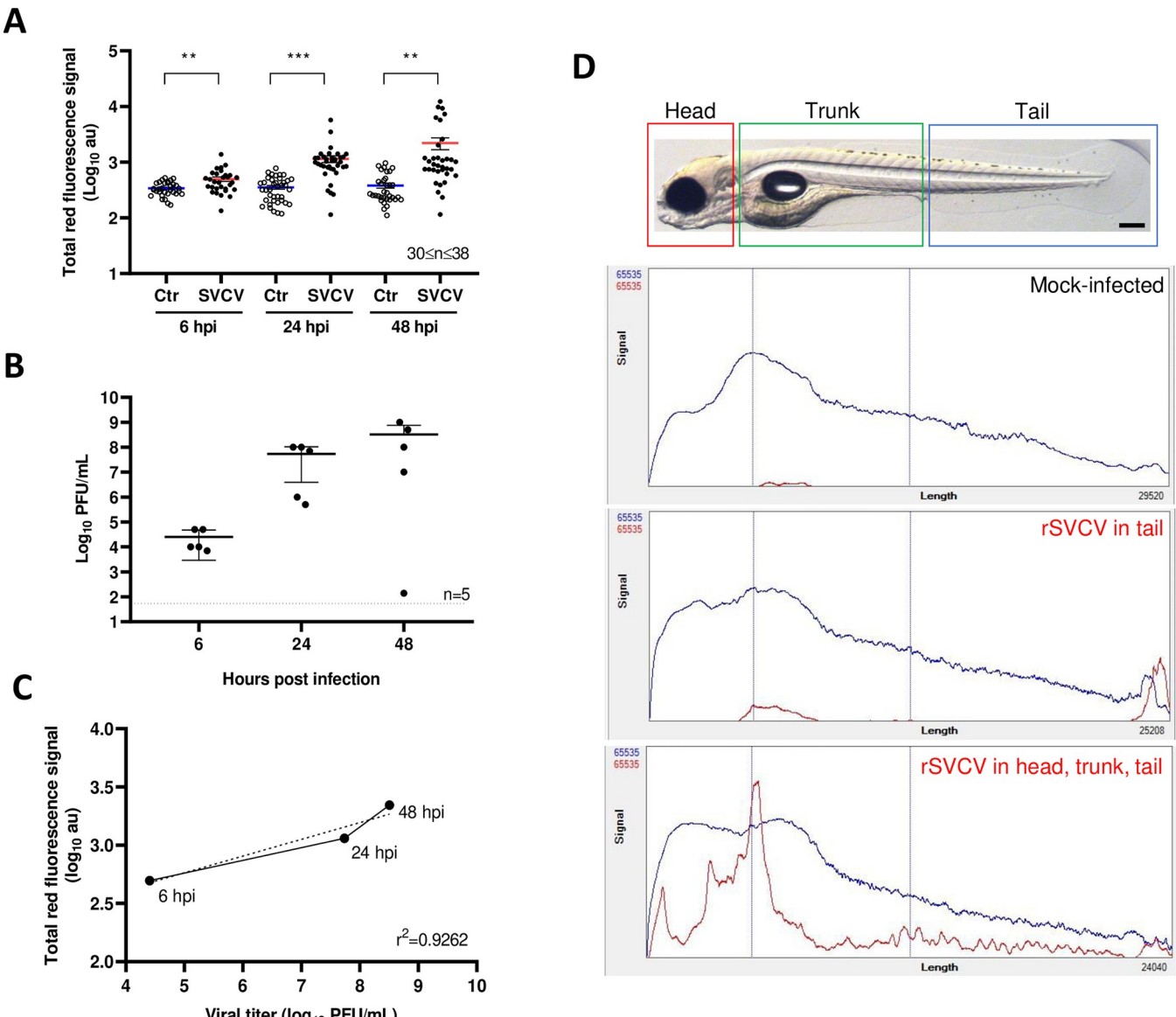

**Fig 8. Quantification of fluorescence intensity in rSVCV-Cherry infected larvae using the COPAS system.** A. Total red fluorescence signal of rSVCV-mCherry infected and mock-infected larvae measured with COPAS system at different times of infection (mean of 30–39 larvae with standard errors). Statistically significant differences are displayed as follows: ***, p value < 0.001; **, p value < 0.01. B. Virus load in larvae analyzed with the COPAS. After passing through the COPAS system, the larvae were harvested in 5 groups of 6 larvae, and virus load was determined by plaque titration in EPC cells. Means are shown together with standard errors. C. Correlation between the two methods: titration in EPC cells and COPAS analysis in larvae infected with rSVCV-mCherry. D. Profiles of mCherry distribution obtained with the COPAS system and segmented as in Fig 7. Examples of mock-infected and rSVCV-mCherry infected larvae profiles at 24 hpi are shown. Blue line: optical density of the larvae, Red line: profile of red fluorescence signal. Scale bar: 200 μm.

and 48 hpi and dot plots of the red fluorescence signal of mock-infected versus rSVCV-mCherry infected larvae were recorded (S9 Fig). The technique was sensitive enough to detect mCherry fluorescence specifically in the infected fish as early as 6 hpi, with intensities above the background signal produced by the embryonic yolk in the mock-infected fish (Fig 8A). We further observed a continuous increase of mCherry fluorescence in infected fish up to 48 hpi. Taking advantage of the absence of any noticeable damage to the embryos during this COPAS analysis, we combined it with plaque assay experiments performed on whole fish sampled after fluorescence reading (Fig 8B). Viral titration showed good correlation with the fluorescence

signals in pools of embryos infected with rSVCV-mCherry ($r^2$ = 0.9262) (Fig 8C). The Profiler software allows analysis and representation of individual fish for the extinction and fluorescence channels. Fish profiles thus show the intensity and distribution of the fluorescence signals (y axis) over the length of the fish body (x axis established from the extension parameter) (Fig 8D).

## SVCV induces a pro-inflammatory response, specific to the route of infection

To investigate the host response to SVCV infection, zebrafish larvae were sampled from 0 to 48 hpi and processed to measure the expression of markers of proinflammatory (*il1b* and *tnfa*) (Fig 9A and 9B) and type I interferon signaling pathways (*ifnphi1* and *isg15*) (Fig 9C and 9D) by RT-qPCR. The infection, when performed by fish immersion in the viral suspension induces the expression of *il1b* from 24 to 48 hpi (Fig 9A) in the absence of induction of type-I *ifn* and *isg15* gene expression (Fig 9C). In contrast, the injection of the viral suspension triggers a more potent proinflammatory response characterized by a significant induction of both *il1b* and *tnfa* (Fig 9B), associated with the activation of interferon signaling as shown by the upregulation of *isg15* expression at late stages of infection (Fig 9D). We next investigated the impact of SVCV infection on macrophages and neutrophils, innate immune cells with diverse roles in inflammatory contexts. Hence, RT-qPCR was carried out to measure the expression of the *mpeg* and *mpx* markers, specific of the macrophages and neutrophil populations, respectively (Fig 9E and 9F). The immersion route of infection induced downregulation of *mpeg* expression at late stages of disease (48 hpi) without modulation of *mpx* (Fig 9E), a phenotype similarly observed in fish infected by IV injection at 24 hpi but to a greater extent for the latter condition (Fig 9F).

## SVCV promotes the recruitment of neutrophils and induces tissue damage

Taking advantage of zebrafish transparency, we next assessed the host response by imaging of fluorescent transgenic lines to characterize the dynamics of immune cells present at the embryonic stage. Thus, challenging zebrafish larvae obtained from transgenic lines expressing GFP markers in macrophages *Tg(mpeg1:eGFP)gl22* or in neutrophils *Tg(mpx:GFP)$^{i114}$*, we evaluated the cell fate and dynamics of these leukocytes during infection. Confocal images of the *Tg(mpeg1:eGFP)gl22* infected larvae showed the recruitment of GFP-positive macrophages in the mCherry-positive infectious foci detected at the level of the head, the trunk or the tail fin at 24 hpi (S10A Fig). Interestingly, infection triggers a change in macrophage morphology visible at the caudal hematopoietic tissue, the trunk and the yolk, whereby the typical dendritic morphology of macrophages in the control condition shifted to a round morphology in rSVCV-infected fish (S10B Fig). At the same time, observation of the *Tg(mpx:GFP)$^{i114}$* larvae enabled the detection of GFP-positive neutrophils at the level of the mCherry positive infection sites (S10C Fig).

Neutrophils play a critical role in the initial defense response against pathogens (phagocytosis of microbes, secretion of granule proteins and other antimicrobial peptides, production of reactive oxygen species, and release of neutrophil extracellular traps) and mediate the proinflammatory response to infection by releasing cytokines. Hence, to investigate the role of these cells during infection, we analyzed the dynamics of their recruitment from the early stages of infection focusing on fish larvae presenting mCherry-positive cells at the tail fin, which constituted the most common phenotype (Fig 10A). Our data showed that at 24 hpi, only a few and clustered mCherry-positive cells were detected in the tail fin with an absence of GFP-positive neutrophils in their close proximity (Fig 10B and 10D). At 48 hpi, we observed a significant increase in the size of the infectious foci resulting from the dissemination of the virus into the caudal fin tissue and the spread to the vascular network. At this later time of infection,

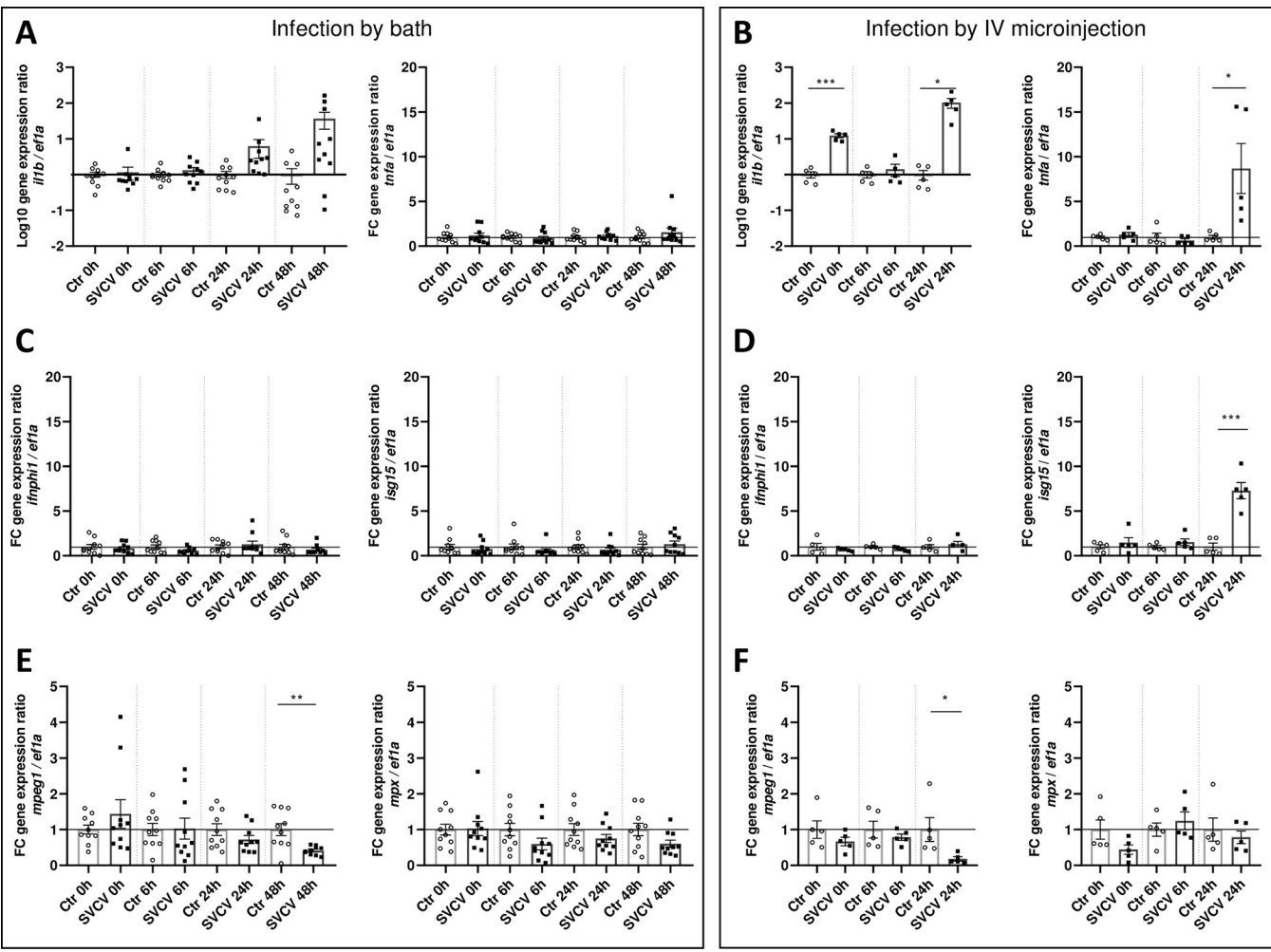

**Fig 9. Analysis of the innate immune response in 3 dpf larvae infected by bath or microinjection in the duct of Cuvier with rSVCV-mCherry.** Expression analysis of the principal genes involved in the anti-inflammatory response (*il1b*, *tnfa*), antiviral response (*ifnphi1*, *isg15*), and marker genes of the two major innate immune cells (*mpeg1* for macrophages, and *mpx* for neutrophils) measured by RT-qPCR when larvae were infected by immersion in the viral suspension (A, C, and E, respectively) or by IV microinjection (B, D, and F, respectively). 5 groups of 5 larvae each were sampled for rSVCV- and mock-infected larvae at 0, 6, 24, and 48 hpi and repeated twice for the bath infection experiment. 5 groups of 5 larvae each of IV rSVCV-infected and non-infected larvae were sampled at 0, 6, and 24 hpi. Each sample was normalized to the reference gene *ef1a*. The normalized expression values were standardized against their respective controls (Control fold change (FC) = 1, basal line). The graphs represent the mean of the fold changes ± SEM of the biological replicates, except for the *il1b* gene which is represented as the $\log_{10}$ of the fold change expression ratio. Statistically significant differences are displayed as follows: ***, p value < 0.001; **, p value < 0.01; *, p value < 0.05.

neutrophils can be detected in the mCherry-positive area (Fig 10C and 10D). Interestingly, transmitted light stereomicroscopic observations conducted at 24 hpi showed that 80% of the infected larvae presented tissue damage at the tail fin in contrast to mock-infected larvae, handled using the same conditions (S11 Fig). Imaging of larvae before infectious challenge rules out the possibility that tissue damage was induced by handling the fish, demonstrating a virally-induced damaging process appearing at 24 hpi.

## SVCV infection triggers a regionalized IL1β expression

Bath immersion of zebrafish larvae in rSVCV inoculum induced a pro-inflammatory response marked by the induction of *il1b* expression measured by RT-qPCR (Fig 9A). To go further in the description of this response, we infected *Tg(il1β:GFP-F)* transgenic larvae expressing GFP

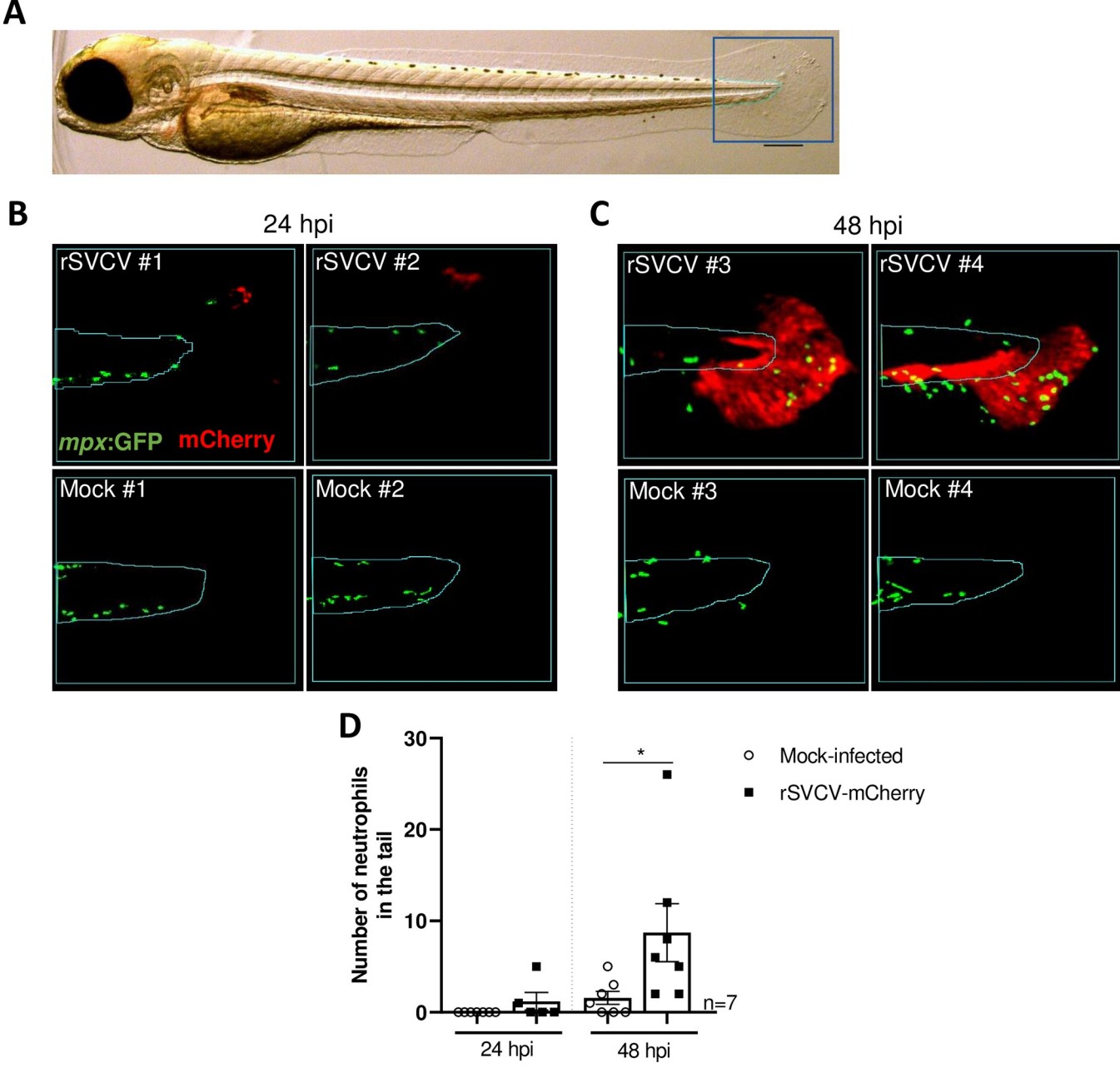

**Fig 10. Neutrophil recruitment to the initial SVCV replication site in the caudal fin.** A. Example of ROI delimitation done for all the analyzed larvae. Scale bar: 200 μm. B. Micrographs of merged GFP (MPX, neutrophils) and mCherry (rSVCV) signals of ROI delimited areas in SVCV-infected and mock-infected larvae at 24 hpi. C. Micrographs of merged GFP (MPX, neutrophils) and mCherry (rSVCV) signals of ROI delimited areas in SVCV-infected and mock-infected larvae at 48 hpi. D. Representation of the number of neutrophils present in the fin fold of the caudal fin of SVCV-infected and non-infected larvae at 24 and 48 hpi. The graph represents the mean ± SEM. Statistically significant differences are displayed as *, p value < 0.05.

under the control of the IL1β promoter enabling the visualization of IL1β producing cells. Uninfected *Tg(il1β:GFP-F)* expressed GFP in keratinocytes at the tip of the caudal fin, in the retina, and neuromasts (Fig 11A–11C) [27]. From 24 hpi, SVCV infection triggered an increase of GFP expression at the tail fin infection site (mCherry-positive) as observed by confocal microscopy (Fig 11A). Higher magnification of these regions showed that the mCherry-positive cells were also GFP-positive suggesting that IL1β was produced by the infected cells.

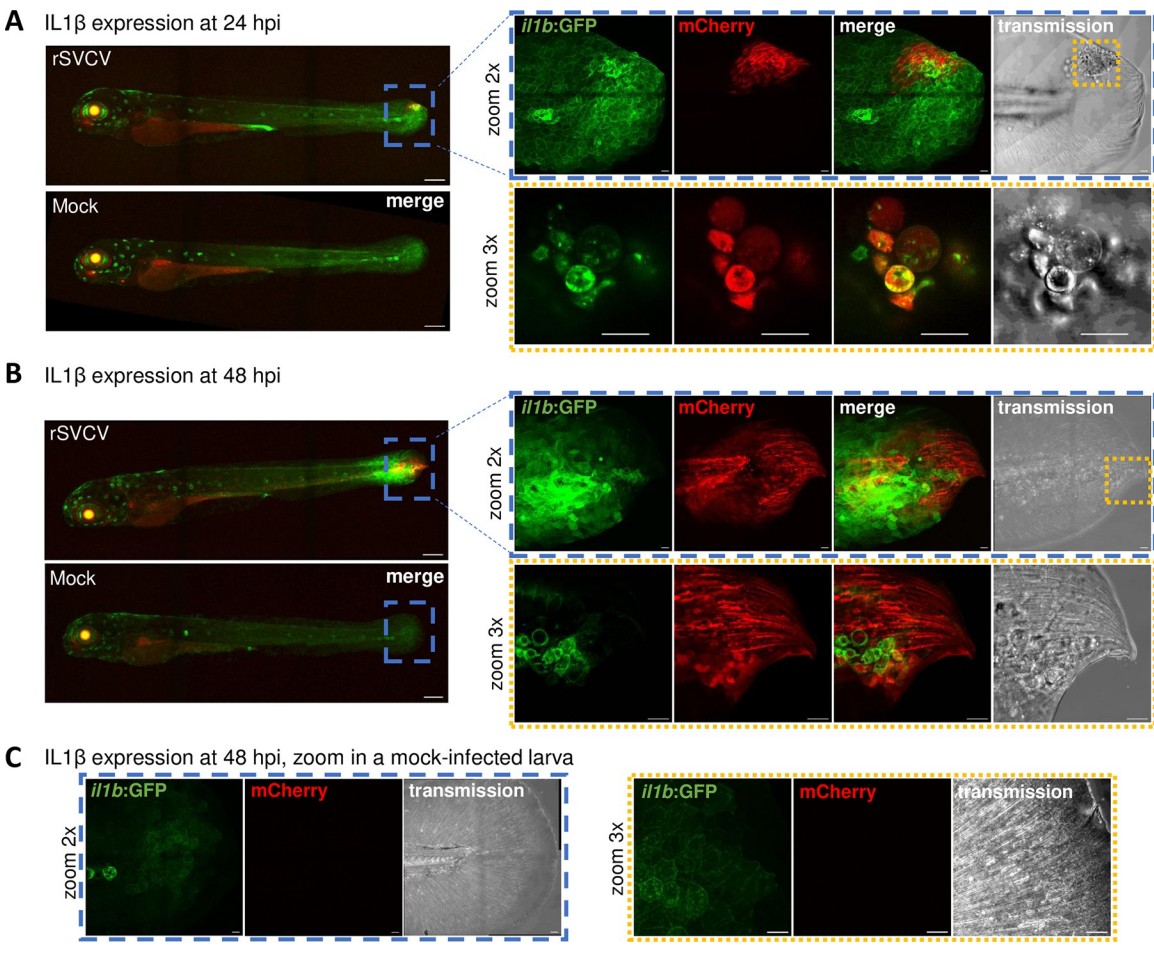

**Fig 11. Visualization of IL1β expression in SVCV-infected larvae.** IL1β expression in mock-infected and SVCV-infected larvae at 24 hpi (A) and 48 hpi (B-C). Confocal images of GFP (IL1β, green) and mCherry (rSVCV, red) signals observed in entire *Tg(il1β:GFP-F)* larvae were taken under a 10× objective. Scale bars: 200 μm. The blue box shows the magnified area of the caudal fin of SVCV-infected (A-B) and mock-infected (C) larvae taken with a 25× objective and zoom 2. Scale bars: 200 μm. The yellow box shows the magnified area of the caudal fin wound taken with a 25× objective and zoom 3. Scale bars: 20 μm.

S2 Video clearly show the infection of keratinocytes, which express GFP in the il1b lineage and are identifiable on the basis of their typical polygonal morphology. This phenotype confirms the results obtained in the target species of carp epithelial cells infection at the caudal fin margin (S4A and S4B Fig and S1 Video). In addition, we observed the swelling of these GFP- and mCherry-positive cells leading to cell lysis and tissue damage (Fig 11A and 11B, and S2 and S3 Videos). In line with the RT-qPCR data (Fig 9), we observed an increase and spread of il1β:GFP-positive signals in caudal fin keratinocytes where the viral infection was progressing (Fig 11B). il1β:GFP positive signals were also observed in hypochord and floor plate located under and above the notochord respectively (S2 and S3 Videos).

## SVCV modulates neutrophilic responses to photoablation

To unravel the role of neutrophils in wound healing in infected and naïve conditions, we established a localized tissue injury model based on infrared laser-targeted photoablation. Transgenic larvae obtained from the *Tg(mpx:GFP)^{i114}* line were anesthetized and mounted under a 2-photon microscope to perform a targeted photoablation (PA) generating a crescent-shaped

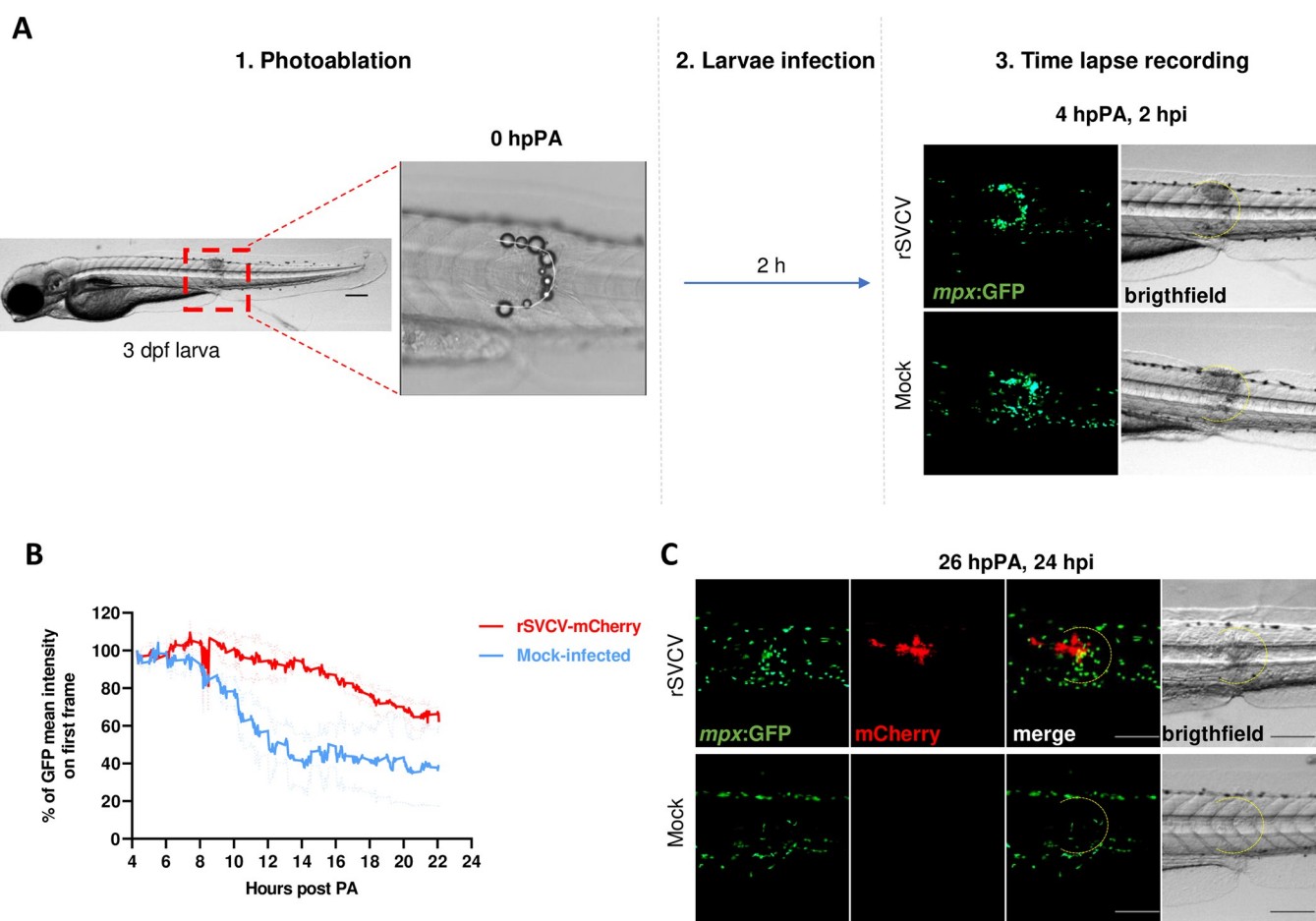

**Fig 12. Neutrophil trafficking in induced injuries.** A. Experimental design. 1. Photoablation (PA): Localization of the injury done by photoablation in 3 dpf larvae and zoom of the dorsal area at the level of urogenital aperture at the moment of PA (0 hour post PA, 0 hpPA). Scale bar: 200 μm. 2. Larvae infection: larvae were rSVCV- or mock-infected by immersion for 1 h, rinsed, and then incubated for 2 h before recording. 3. Time-lapse recording: Images taken with the stereomicroscope during the mounting of the larvae in agarose just before starting the recording of the time-lapse (4 hpPA, 2 hpi) of rSVCV- or mock-infected larvae. Images of the GFP signal of the neutrophils in green, and photoablation area marked with a yellow dotted crescent in the brightfield channel. B. Representation of the percentage of GFP mean intensity on the first frame of the time-lapse record in the photoablation area of rSVCV- or mock-infected larvae from 4 hpPA and 2 hpi to 22 hpPA and 20 hpi. The graph represents the mean ± SEM. C. Stereomicroscope images of GFP (*mpx*, neutrophils) and mCherry (rSVCV) signals of photoablation site of rSVCV- or mock-infected larvae at 26 hpPA and 24 hpi. The photoablation area is marked with a yellow dotted crescent. Scale bars: 200 μm.

injury (Fig 12A). Immediately after PA fish were recovered and distributed in two groups incubated in a viral suspension (rSVCV) or in control medium (mock infected) for an hour. After virus removal, fish were incubated for 2 additional hours before monitoring of neutrophil recruitment at the site of tissue injury by combining dynamic confocal acquisition with real-time stereomicroscopy. As expected, 4 hpPA, neutrophils were already present at the site of the injury in mock- and SVCV-infected larvae (2 hpi) (Fig 12A).

Quantitation of the neutrophil recruitment was assessed by analyzing the GFP intensity on confocal dynamic acquisitions (Fig 12B and S4–S7 Videos). In mock-infected larvae, GFP intensity did not increase after the start of the monitoring indicating that the peak of neutrophil recruitment had already been attained at 4 hpPA. From 8 to 14 hpPA, mock-infected fish showed a decrease in GFP intensity, which subsequently reached a plateau suggesting a resolution of inflammation (Fig 12B). In contrast, rSVCV-infected fish showed a persistence of neutrophils at the site of injury up to 8 hpPA, before the occurrence of the decrease in GFP

intensity, without ever reaching a plateau as observed in mock-infected larvae (Fig 12B). Hence, our results demonstrate that in a model of PA-induced tissue injury, SVCV infection interferes with the resolution of the inflammatory response, which prevents a return to homeostasis. Images acquired in parallel with the stereomicroscope at 26 hpPA corroborated these dynamic results. In naïve fish, neutrophils were no longer detected at the site of PA, in contrast with rSVCV-infected larvae where neutrophils remained at the site of injury in the presence of mCherry-positive cells (Fig 12C and larva rSVCV #1, 2, and 3 of S12 Fig). Interestingly, when infection occurred outside the area of injury (larva rSVCV #4, 5, and 6 of S12 Fig), neutrophils were not detected in the crescent but at the SVCV infection foci, indicating that their persistence is based on a regionalized virally-induced signaling rather than a systemic response.

## Discussion

Although several fish viruses have been extensively studied, there remains limited knowledge on the route of aquatic virus entry into their hosts. It has been experimentally demonstrated that SVCV transmission occurs through contaminated water by direct excretion of the virus from infected fish and does not necessarily require any vector [9]. It is commonly believed that viruses infect fish through the gills and SVCV is no exception [9,28]. However, this postulate has been challenged by studies based on innovative technologies showing that instead fins and wounds on the skin constitute the main portals of entry for two related rhabdoviruses, Infectious Hematopoietic Necrosis Virus (IHNV) [21] and Viral Hemorrhagic Septicemia Virus (VHSV) [24]. Indeed, recombinant, traceable viruses expressing reporter genes such as luciferase were generated enabling highly sensitive monitoring of initial sites of infection at the whole fish level by live imaging. Thus, trout infected by bath immersion with recombinant virus encoding luciferase and monitored by noninvasive bioluminescence assays provide direct evidence that the fins were the primary sites of IHNV and VHSV replication [21,24]. In the present study, we describe the establishment of a reverse genetics system for SVCV allowing *in vivo* investigation of viral entry and spreading by advanced phenotyping approaches. A recombinant SVCV expressing the Akaluciferase [22] was engineered and a noninvasive bioluminescence assay was developed in carp using a luciferin analog, Akalumine-HCL, generating near-infrared bioluminescence [23]. In this assay, the AkaLUC substrate is directly added to the water of the tank housing carp, diffusing into the fish body thanks to its high solubility and deliverability. This is a notable technical advance as compared to other modes of administration of luciferin which typically requires injection. This also contributes to the refinement of the experimental procedures improving animal welfare. Bioluminescence and fluorescence imaging of juvenile carp at different times post infection with the rSVCV allowed us to detect the first SVCV replication sites at superficial levels of the fins and of the skin, which suggests that they are the major portal of entry for SVCV in carp. Cellular imaging further indicates that epithelial cells constitute the targeted cells of the virus in waterborne infection (S4A and S4B Fig and S1 Video). At the same time, no bioluminescent signals were observed in the gills (Fig 4A). The fins, and more precisely the caudal fin, are the primary sites of viral replication and appear to be the most permissive organs for several viral infections at early stages after bath immersion, including novirhabdoviruses in trout [21,24] and cyprinid herpesviruses in carp and goldfish [20,29]. It has been demonstrated that mucus removal and epidermal lesions clearly enhance cyprinid herpesvirus entry in carp [30]. These lesions may be caused by physical contact and/or fin nipping between fish. Here, we demonstrated that artificially-induced lesions in the caudal fins favor SVCV entry in carp reinforcing the hypothesis of lesion-dependent entry of SVCV (Fig 4B).

To further characterize SVCV disease progression, we investigated SVCV spreading mechanisms and host responses in a zebrafish model using a traceable fluorescent rSVCV-mCherry (Figs 6 and 7). We took advantage of the small size and optical transparency of zebrafish larvae to determine the virus entry and spreading assessed by fluorescence microscopy. The establishment of this reliable zebrafish model is based on fish immersion in rSVCV inoculum, similar to the situation of carp infection in the field. Short-term exposure to rSVCV by bath immersion was sufficient to initiate infection. Initial mCherry fluorescence signals were detected in superficial tissues localized in different parts of the body but mainly targeted the caudal fin (in 60% of the larvae) as observed in its natural host. Remarkably, at early time points no replication sites could be detected in the region of the gills. It should be noted that the formation of the gills in zebrafish larvae starts at 3 dpf (the development stage of larvae used in SVCV infection experiments), they lack respiratory lamellae and do not become fully functional until 14 dpf [31]. However, larvae exhibit patterned operculum movements at 3 dpf, rendering direct contact with SVCV possible but virus infection was never observed. Interestingly, infectious challenge of juvenile zebrafish by immersion also led to the detection of rSVCV-mCherry positive signals in the fins, which constitute the initial sites of SVCV replication in a high proportion of infected fish (S7C Fig). Thus, the early localization of mCherry or bioluminescent signals in the epithelial cells of carp and zebrafish fins (at different stages of development) indicates that these cells located in the superficial layers of the peripheral organs are highly permissive to SVCV infection and constitute the major portal of entry of SVCV in its host (S4A and S4B Fig, S1 and S2 Videos). Finally, our results obtained in carp with amputated caudal fins or zebrafish with photoablation tissue injury (Figs 12 and S12), reinforce the hypothesis of lesion-dependent entry of SVCV in its host, a scenario likely encountered in semi-intensive farming practices.

Our model of SVCV infection based on immersion of zebrafish larvae in viral inoculum, led to a high rate of fish death (80%) by 3 days post infection. This contrasts with previously established models showing that early stages of zebrafish development (from 2 cells to 3 dpf) were resistant to SVCV infection when challenged by immersion [28]. The authors suggested that the stage at which zebrafish becomes susceptible to SVCV (5 dpf) in water correlates with the onset of respiratory movements following the opening of the mouth and gill slits, suggesting that SVCV may infect zebrafish larvae via the gills. However, intravenous injection of SVCV in 3 dpf larvae triggers robust infection. Moreover, the presence of the virus was only monitored by RT-qPCR performed on whole larvae, hence reducing the chance of detecting abortive initial sites of virus. In contrast, López-Muñoz and colleagues successfully established SVCV infection of 3 dpf larvae by immersion using a different viral strain, a higher temperature of incubation (from 24 to 26°C), and a longer incubation time (from overnight to 24 hours) [32]. In our model, the incubation time has been successfully shortened to 1 hour with a temperature of incubation of 24°C. To conduct the fluorescence imaging of infected larvae in optimal conditions of observation, we carried out the experiment on nacre zebrafish mutants, as they are natively transparent by the lack of melanophores. Nacre mutants harbor a mutation in the *mitfa* gene, which regulates neural crest-derived pigments, and results in a loss of melanoblast and mature melanocytes [33]. The impact of zebrafish genetic background on susceptibility to pathogen infection has not been extensively investigated. Levraud and colleagues demonstrated that *nacre* mutants were as susceptible as wild-type zebrafish to *Listeria monocytogenes* infection [34]. Recent studies reported the generation of a *mitfa* knock-out zebrafish line using the CRISPR/Cas9 system to unravel the impact of MITFA depletion on fish immune responses, although in a different phenotype and genotype than that of the nacre mutant [35]. Bian and colleagues described a significantly higher number of macrophages and higher transcription levels of *tlr2*, *tlr4*, *nod2*, *myd88*, *nfkb1*, *nfkb2*, *rela*, and *relb* as compared to

wild-type fish after exposure to *Edwardsiella tarda*. In contrast, Chen and colleagues, using a distinct *mitfa* knockout line, showed a transcriptional down-regulation of *tlr4a*, *tlr4b*, *tlr2*, *lyz*, *il10*, *tnfa*, *Ca3*, *il1b and il10* genes in *mitfa* knockout fish suggesting impaired innate immune responses [36]. Interestingly, the expression of the pro-inflammatory cytokines *il1b* and *il6* was decreased at 1 dpf, but increased at 20 dpf in *mitfa*$^{-/-}$ larvae suggesting that at the molecular level, the absence of MITFA can modulate the antibacterial response depending on the larval stage of development [35]. However, virus microinjection in the yolk of 3 dpf larvae led to comparable levels of human norovirus replication in both nacre and wild-type larvae [37]. Further studies will be necessary to evaluate the effect of MITFA mutation and/or depletion on fish responses to viral challenges.

Zebrafish larvae have been highly beneficial for the study and evaluation of the fish antiviral innate immune responses [38,39] and for characterizing the interferon signaling pathway [28,40]. In the context of SVCV infection, host immune responses are highly dependent on the mode of virus administration. Intravenous injection of SVCV in the caudal hematopoietic tissue triggers the activation of *ifnphi1*, interferon-stimulated genes (*mxa*, *mxb* and *viperin*) and *tnfa* [28]. These results were confirmed in another model of SVCV intravenous injection in the duct of Cuvier [26]. The virus similarly induced interferon signaling (*ifnphi1* and *2*, *mxab*, *rarres3*, *ifit13a*, and *ifit17a*), in addition to upregulation of proinflammatory cytokine genes involved in the inflammatory response (*il1b*, *tnfa*, and *il6*) and markers of the inflammasome (*asc*, *caspa*). In contrast, López-Muñoz and colleagues showed that zebrafish larvae are unable to mount an interferon antiviral response against waterborne infection by SVCV [32]. However, infection resulted in the induction of *il1b* and, to some extent, of *tnfa* and lymphotoxin a (*lta*). In the present study, waterborne infection also failed to elicit the expression of genes involved in the interferon signaling pathway but rather induced a significant IL1β expression. Interestingly, we were able to combine RT-qPCR approaches to fluorescence imaging to investigate the tissue specificity of this response. Infection of *Tg(il1β:GFP-F)* transgenic larvae allowed the visualization of IL1β expressing cells (Fig 11). 3D confocal imaging at cellular resolution further demonstrated that keratinocytes, which express GFP in the IL1β lineage and are identifiable on the basis of their typical polygonal morphology [27], are positive rSVCV-mCherry-infected cells. During infection, the green fluorescence intensity of the *Tg(il1β:GFP-F)* increased at the site of virus replication before spreading to the connected tissues. Concomitantly, we observed swelling and death of mCherry-positive and IL1β-expressing cells, which might depend on inflammasome activation as reported by Varela and colleagues [26], although in a distinct model of intravenous infection of SVCV.

Thus, the comparison of intravenous and waterborne infections in our study reveals the link between the mode of administration and the host responses (Figs 7 and S8 Fig). Taking advantage of recombinant SVCV, we first investigated the impact of the infection route on virus tropism in whole fish. Immersion favors entry at the level of superficial tissues, before reaching the vascular network and target organs of the virus (heart, liver, head kidney, and intestine). In contrast, upon intravenous infection in the duct of Cuvier, the virus infects and spreads in the cells of the vascular network before spreading to the whole organism. This shortened the survival of the infected animals without strong modification of the mCherry expression pattern at late stages of infection.

As previously demonstrated [26], we showed that SVCV infection triggers a reduction of the expression of macrophage marker (Fig 9E and 9F). We further characterized the impact of SVCV on neutrophil behavior, which plays a major role in local immune responses when activated after tissue injury, ensuring tissue regeneration. First, the SVCV infection of *Tg(mpx:GFP)*$^{i114}$ transgenic larvae demonstrates the active and progressive recruitment of neutrophils at the site of infection (Figs S10 and Fig 10), when localized in the tail fin, superficial tissues of

the trunk or the head. Notably, the recruitment of neutrophils to the infection sites in the cau- dal fin appears delayed compared to other primary sites of viral replication Figs S10 and 10), which underscores the importance of fins as primary sites of infection for many fish viruses. Further investigations will be necessary to understand the particular permissiveness of the fins to virus infections. We also evaluated the immune cell behavior and dynamics in a photoabla- tion model of tissue injury. As reported, upon local injury in fish, neutrophils were attracted by a $H_2O_2$ gradient of dying epithelial cells to clear cellular debris [41], protecting against envi- ronmental pathogens and controlling the inflammatory response [42]. Three to four hours after injury, neutrophils were shown to leave the site of injury allowing tissue healing/wound repair and regeneration [42]. In infected fish, we similarly observed neutrophil recruitment at the PA site (36 hpPA) (Figs 12 and S12 and S4–S7 Videos) but their persistence up to 8 hpPA suggests that SVCV infection interferes with the resolution of the inflammatory response and prevents a return to homeostasis. Interestingly, when infection occurred outside the PA site, neutrophils persistence is only observed at the level of the SVCV replication site suggesting a local virally-induced immune response rather than systemic inflammation.

Overall, we established a novel zebrafish model of waterborne SVCV infection which can be used for antiviral screening. We demonstrated that rSVCV-mCherry fluorescence signals were correlated with viral titer, enabling detailed spatio-temporal monitoring of the infection process in live zebrafish larvae by combining fluorescence microscopy and larvae-adapted flow cytome- try. This highly sensitive method enabled the rapid and non-invasive analysis of rSVCV- mCherry fluorescence signals at high throughput. Data processing enabled us to locate and quantify the fluorescent signal during infection (Fig 8). Hence, our model will be amenable to test the efficacy of antiviral, anti-inflammatory, or immunostimulants for the control of SVCV infection. Our integrated study combines *in vitro*, zebrafish, and carp models as target species, highlighting the value of model organisms to mimic and predict virus-induced diseases. Thus, it opens up new avenues for *in vivo* studies of emerging viral fish diseases challenging the sustain- ability of aquaculture and paves the way for the identification of novel drug screening strategies.

## Materials and methods

### Ethics statements

All animal studies were carried out in strict accordance with European guidelines and recom- mendations on animal experimentation and welfare (European Union Directive 2010/63). All animal experimental procedures were approved by the local ethics committee on animal experimentation (Comité d'éthique appliqué à l'expérimentation animale pour le centre de recherches INRAE de Jouy-en-Josas; COMETHEA INRAE no. 45) and were authorized by the Ministère de l'Éducation nationale, de l'Enseignement supérieur et de la Recherche under the numbers: APAFIS#29801–2021021110262075 v2 as well as fish facilities (authorization num- ber C78-720). To minimize animal suffering and distress, all manipulations were carried out under light anesthesia. Anesthesia was performed by bath immersion with tricaine (0.005%). A lethal challenge with SVCV typically results in acute disease characterized by anemia and punctiform hemorrhages. Therefore, fish were monitored twice daily for clinical signs and sur- vival. Upon display of typical infection symptoms, animals were humanely euthanized by bath immersion using a lethal dose of tricaine (MS-222 Sigma at 0.015%).

### Whole-genome SVCV phylogenetic analysis

To generate a phylogenetic tree of SVCV strains, the complete genomic nucleotide sequences of 16 SVCV isolates including the Fijan strain (Genbank # AJ318079.1) which was used as the basis for the reverse genetics system described here and the sequence for Pike fry rhabdovirus

F4 were downloaded from NCBI Genbank. The accession numbers and associated metadata (host species date of sampling and country of origin) for these sequences are summarized in Table 1. A whole-genome nucleotide alignment of these sequences was performed using MUS-CLE [43] in Seaview (version 5) software [44]. The alignment was then assessed using Models Analysis in MEGA X (version 10) for the choice of substitution model for downstream Maximum-Likelihood phylogenetic analysis [45]. The substitution model with the lowest Bayesian Information Criterion (BIC) score was the General Time Reversible model with discrete Gamma distribution (GTR+G) and was subsequently chosen for the phylogenetic tree analysis. The Maximum Likelihood tree was built using PhyML [46] with 4 substitution rate classes and 1000 bootstrap replicates. The tree was drawn using FigTree 1.4.4 (http://tree.bio.ed.ac.uk/software/figtree/) and rooted using Pike fry rhabdovirus. Branch lengths are measured in number of substitutions per site.

## Cells and virus

*Epithelioma Papulosum Cyprini* (EPC) cells were maintained at 25°C in GMEM/HEPES 25 mM medium supplemented with 2 mM L-glutamine (Eurobio) and with 10% fetal bovine serum (FBS) (Clinisciences) [47,48]. SVCV Fijan strain and rSVCV were propagated in monolayer cultures of EPC cells at 25°C as previously described [49,50]. Virus titers were determined by plaque assays on EPC cells under an agarose overlay (0.35% agarose in Glasgow's modified Eagle's medium with 25 mM HEPES supplemented with 2% FBS and 2 mM L-glutamine). At 2 to 3 days post infection, cell monolayers were fixed with 10% formalin and stained with crystal violet. BSR T7/5 cells are baby hamster kidney 21 (BHK-21) cells that constitutively express T7 RNA polymerase [18] and were kindly provided by Dr. Karl-Klaus Conzelmann. They were maintained in Dulbecco's modified Eagle's medium supplemented with 10% FBS, 2 mM L-glutamine, and 500 g/mL of G418 Geneticin (Thermo Fisher Scientific) every two passages.

## Plasmid constructs and recombinant virus recovery

A full-length cDNA copy of the SVCV RNA genome was constructed by assembling five overlapping cDNA fragments generated through reverse transcription (RT)-PCR as depicted in S1 Fig. The pairs of primers used were NAET7SVC/KPNSVC for fragment 1, KPXHSVC/XHOSVC for fragment 2, XHSMSVC/SMASVC for fragment 3, and SMEAGSVC/EAGSVC for fragment 4 (Table 2). Fragment 5 was generated with primers EAGRIBSVC/T7TERMSAC from a previous pIHNV plasmid DNA [51].

The 5' end of fragment 1 (2,037 nt) consists of a *NaeI* restriction enzyme (RE) site, a promoter sequence for the T7 RNA polymerase, three G residues to enhance promoter activity and is fused to the first nucleotide of the SVCV antigenomic sequence. Fragment 1 covers the N gene and a large part of the P gene up to a naturally occurring *KpnI* RE site. Fragment 2 (1,050 nt) and Fragment 3 (1,569 nt) introduce a *XhoI* (at positions 3064–3069 of the genome; GCTGAG -> CTCGAG) and a *SmaI* (at positions 4631–4636 of the genome; CAGATT -> CCCGGG) RE sites in the intergenic regions between the M and G genes and the G and L genes, respectively. Fragment 4 (6,343 nt) comprises the complete L gene and includes an *EagI* RE site at the 3' end (at position 10976–10981 of the genome; GGGGGA -> CGGCCG). Finally, fragment 5 (213 nt) contains the *EagI* RE site with the trailer sequence fused to the sequence of the hepatitis delta virus antigenomic ribozyme, followed by a T7 terminator sequence, and a *SacI* RE site. All fragments were cloned in a pJET1.2 cloning vector, and the sequences were verified by Sanger sequencing. Fragments 1 to 5 were sequentially cloned into a pBluescript SK- vector to obtain the complete SVCV antigenome construct, named pSVCV (Fig 1B). This genome differs from the genome of the Fijan reference strain (Genbank #

AJ318079.1; [15]) by the three additional RE sites described above and three nucleotide substitutions, two of which are synonymous mutations and are located in the N (T->G at position 1284) and G (A->G at position 3897) genes and one is a missense mutation in the G gene (A->G at position 4354) leading to a threonine to alanine substitution.

The intergenic regions within the SVCV Fijan strain genome were aligned and compared (S2A Fig) to obtain the minimal consensus sequence DDDRTATGAAAAAAACTAACAGASATCATG (with D = G, A or T, R = G or A, and S = G or C). This minimal sequence was used in all expression cassettes described below for the insertion of additional genes in the SVCV genome (S2A Fig).

PCR fragments containing the coding region for GFPmax, mCherry, ffLUC, and akaLUC proteins and flanked by the GS and GE transcription signals of SVCV (Fig 2) were obtained using primers in Table 2. These final PCR products were digested with *SmaI* or *XhoI* and inserted into a *XhoI*- or *SmaI*-linearized pSVCV construct, leading to pSVCV-Cherry M/G, pSVCV-Cherry G/L, pSVCV-ffLUC M/G, pSVCV-ffLUC G/L, and pSVCV-akaLUC G/L (Fig 2A). A pSVCV with two reporter genes was constructed by the insertion of GFPmax into the *XhoI* site and the mCherry gene into the *SmaI* site leading to pSVCV GFPmaxCherry (Fig 2A). For the insertion of the *SmaI*-PCR products into the M-G intergenic region, the plasmid was treated by the Klenow fragment to obtain compatible blunt ends, after *XhoI* digestion.

To generate SVCV helper plasmids for reverse genetics, SVCV N, P, and L genes were amplified by PCR using pSVCV as a template and specific primers (Table 2). Each gene was cloned in the pCITE vector digested by *NcoI* and *XhoI* using the In-Fusion HD Cloning Kit (Clontech).

rSVCV were generated by transfecting BSRT7/5 cells with a mixture of four plasmids by lipofection: 1 genomic plasmid and 3 helper plasmids combined with the transfection reagent Lipofectamine 2000 (Invitrogen). For virus recovery, the day prior to transfection, $1 \times 10^6$ cells/well were seeded in wells of six-well plates so as to reach 90% confluence the following day. On the day of transfection, the cells were maintained in DMEM medium without antibiotics and serum. Cells were transfected with 5 μg of pSVCV plasmid DNA and three helper plasmids: pSVCV-N (2 μg), pSVCV-P (2 μg) and pSVCV-L (1 μg), which encode respectively the nucleoprotein N, the phosphoprotein P, and the RNA-dependent RNA polymerase L, together forming the replicative complex. Plasmids and Lipofectamine (1:3 ratio) were diluted separately in 150 μL of Opti-MEM Medium (Gibco) and diluted DNA was then added to the Lipofectamine mix. The DNA-lipid complex was incubated for 5 minutes at room temperature and was added to cells in a drop-wise manner. After transfection, the cells were incubated for 6 h to 12 h at 37˚C, the transfection mix was then removed and the cells were shifted to 25˚C with GMEM 2% FBS for 7 days. Next, cells were scraped and mixed with the supernatant by scratching wells (the recovered virus at this stage corresponds to passage 0, or P0), clarified by low-speed centrifugation, and were used to infect a monolayer of freshly seeded EPC cells in 24-well plates and incubated at 25˚C (dilution 1:10 in GMEM 2% FBS medium). After 1–4 days, when a complete cytopathic effect was obtained, the supernatant from this culture (supernatant of passage 1, or P1) was collected and used to infect fresh cells (dilution 1:100 in GMEM 2% FBS medium). The supernatant of P2 was finally titered and stored at -80˚C. EPC cells infected with rSVCV expressing mCherry or both GFPmax and mCherry were observed with a UV-light microscope (Carl Zeiss).

## Experimental carp infection and bioluminescence imaging

Virus-free juvenile common carp (*Cyrpinus carpio carpio*, R3×R8 line [52]) were acclimated at 10˚C or 13˚C by progressively reducing the rearing temperature (starting temperature of 25˚C) by increments of 2˚C every 48 h. Fish (50 fish per group; mean weight, 0.52 g or 0.81 g)

were then infected by immersion in tanks filled with 3 L of freshwater with recombinant rSVCV viruses (final titer, $5 \times 10^4$ PFU/mL) for 2 h. Tanks were then filled up to 15 L of freshwater (renewal 5 L/h). Control fish were mock infected with cell culture medium under the same conditions. Mortalities were recorded daily over 115 days.

For *in vivo* bioluminescence imaging, carp were divided into two groups with one group infected by immersion with rSVCV-akaLUC G/L as described above and with the other having their caudal fins cut prior to infection by immersion with rSVCV-akaLUC G/L (mean weight, 1.91 g). At 3, 7, and 24 days post infection, 4 fish were randomly transferred to a small tank with water containing akalumine-HCl substrate (40 mM at 1/10,000 dilution). Two hours later, anesthetized fish were imaged using an IVIS Spectrum BL imaging system (PerkinElmer). Living Image software (version 4.7.3, PerkinElmer) was used to acquire both bioluminescent and photographic images.

## Zebrafish and rSVCV-mCherry infection

Zebrafish (*Danio rerio*) were maintained at 28˚C on a 14 h light/10 h dark cycle. Fish were housed in the animal facility of our laboratory which was built according to the respective local animal welfare standards. In order to conduct fluorescence imaging, transparent fish, *nacre*$^{w2/w2}$ mutants, were produced for the experiments [33]. The lines used in this study were *Tg(mpeg1:eGFP)gl22* [53], *Tg(mpx:GFP)*$^{i1114}$ [54], and *Tg(il1β:GFP-F)* [27].

Nacre zebrafish larvae at 3 days post fertilization (dpf) were infected by bath immersion with rSVCV-mCherry ($2 \times 10^6$ PFU) (40–44 larvae) or mock-infected with cell culture medium (20 larvae) in a final volume of 200 μL and incubated at 24˚C. After 1 h, fish were rinsed in water to remove viral particles still in suspension and incubated at 24˚C in a sterile 1× E3 solution [55] containing 0.0003% methylene blue (Sigma). For the experiments with juveniles, nacre zebrafish larvae at 25 dpf were infected by bath immersion with rSVCV-mCherry ($10^7$ PFU) (20–31 larvae) or mock-infected with cell culture (10 larvae) in a final volume of 100 mL and incubated at 24˚C. After 2 h, up to 1 L of 1× E3 solution was added. Juveniles were fed twice daily and half of the water was renewed daily. Both larvae and juveniles were observed daily for the detection of mCherry fluorescent foci with an MZ10F fluorescent stereomicroscope (Leica Microsystems) and mortality was recorded. Experiments were repeated 3 times for larvae and twice for juveniles.

Infection by microinjection was also tested in nacre zebrafish larvae at 3 dpf. Larvae were anaesthetized with eugenol solution (0.0075%), and placed on a plate containing 1% Phytagel (Sigma-Aldrich) diluted in 1× E3 solution to allow individual microinjection with a glass microneedle using a mechanical micromanipulator (M-152, Narishige) and a Femtojet microinjector (Eppendorf). Larvae were microinjected in the duct of Cuvier with 1 nL of rSVCV-mCherry (40 PFU) or with 1 nL of cell culture for mock-infected larvae. Then, larvae were maintained in 1× E3 solution with 0.0003% methylene blue at 24˚C.

The viral titer was analyzed in both larvae and juveniles at 0, 6, 24, and 48 hours post infection (hpi). Zebrafish were collected individually (10 larvae or 10 juveniles) or pooled by groups of 6 individuals (5 groups) in tubes and kept at -80˚C until titration. Two small sterile ceramic beads were added to each tube with 100 μL of cell culture medium supplemented with gentamicin (Sigma) and samples were homogenized using 2 cycles of 15 s at 6,000 rpm using a Precellys 24 Touch tissue homogenizer (Bertin technologies). Samples were then clarified for 3 min at 10,000 rpm. Lysates were then serially diluted 10-fold in cell culture medium supplemented with 2% FBS and used to infect EPC cells in 96-well plates and incubated at 25˚C. The viral titer was calculated at 48 hpi by detecting fluorescent foci with a Nikon eclipse TE200 fluorescence microscope.

**Table 3. Primers to evaluate zebrafish gene expression and viral replication.**

| Gene | NCBI accession no. | Name primer | Sequence (5' to 3') | References |
| --- | --- | --- | --- | --- |
| il1b | NM_212844.2 | zIL1b-fw | TGGACTTCGCAGCACAAAATG | [58] |
| | | zIL1b.rv | GTTCACTTCACGCTCTTGGATG | |
| tnfa | NM_212859 | zTNFa-fw | TTCACGCTCCATAAGACCCA | [58] |
| | | zTNFa-rv | CCGTAGGATTCAGAAAAGCG | |
| ifnphi1 | NC_007114.7 | zIFNphi1-fw | GAATGGCTTGGCCGATACAGGATA | [59] |
| | | zIFNphi1-rv | TCCTCCACCTTTGACTTGTCCATC | |
| isg15 | NC_007116.7 | zISG15-fw | AGAAGGGCCAGGTCAAAACT | [60] |
| | | zISG15-rv | CGAGCTGTCTGCCTTTGAAA | |
| mpx | NM_212779.2 | zmpx-fw | TCAATATGAGGACGCCGTTTCT | [61] |
| | | zmpx-rv | GAATGCGATTGGAAACCAGTCT | |
| mpeg | NM_212737.1 | zmpeg1-fw | GTGAAAGAGGGTTCTGTTACA | [58] |
| | | zmpeg1-rv | GCCGTAATCAAGTACGAGTT | |
| ef1a | ENSDARG00000020850 | zEF1a-fw | TTCTGTTACCTGGCAAAGGG | [58] |
| | | zEF1a-rv | TTCAGTTTGTCCAACACCCA | |
| Nsvcv | AJ318079.1 | Nsvcv-fw | TGAGGTGAGTGCTGAGGATG | [32] |
| | | Nsvcv-rv | CCATCAGCAAAGTCCGGTAT | |

RT-qPCR was used to evaluate the expression levels of the main immune genes of zebrafish larvae infected or not with rSVCV, which includes *interleukin 1 beta* (*il1b*), *tumor necrosis factor a* (*tnfa*), *interferon phi 1* (*ifnphi1*), and *ISG15 ubiquitin like modifier* (*isg15*), along with the gene expression markers of the main immune cells, *macrophage expressed 1 tandem duplicate 1* (*mpeg1.1*) for macrophages, and *myeloid-specific peroxidase* (*mpx*) for neutrophils (Table 3). Viral replication was confirmed by measuring the relative expression of the gene encoding the SVCV N protein as previously described [32]. Briefly, larvae infected with rSVCV-mCherry or mock-infected larvae were harvested under RNAse-free conditions (5 pools of 5 larvae for each condition) at 0, 6, 24, and 48 hpi. Homogenization of the larvae was carried out as described above, but this time, together with the two sterile ceramic beads, 350 μL of Buffer RTL Plus and 10 μL of β-mercaptoethanol (Sigma). RNA was extracted using the RNeasy Plus Mini Kit (Qiagen) following the manufacturer's protocol. Total RNA for each sample was quantified using a NanoDrop 2000c spectrophotometer (Thermo Scientific) and cDNA was synthesized with the iScript Advanced cDNA Synthesis kit for RT-qPCR (Bio-Rad) using 0.45 μg of total RNA. Finally, qPCRs were conducted in 22-μL reaction volumes using 10 μL of iTaq Universal SYBR Green Supermix (Bio-Rad), 5 μL of ultrapure water, 2.5 μL of each specific primer (2 μM), and 2 μL of cDNA template. All reactions were performed using three technical replicates in a Realplex² Mastercycler epgradient S (Eppendorf), with an initial denaturation step (95°C, 30 s), followed by 40 cycles of denaturation step (95°C, 5 s), and one hybridization-elongation step (60°C, 30 s). The relative expression levels of the targeted genes were normalized using the Pfaffl method [56] and the *eukaryotic translation elongation factor 1 alpha 1 like 2* (*ef1a*) was used as a reference gene.

## Immunohistochemistry and tissue clearing

Zebrafish larvae were fixed in 4% formaldehyde overnight at 4°C. After fixation, larvae were rinsed in PBS-tween 0.1% and permeabilized by incubation in PBS containing 0.2% Triton X-100, 2.3% glycine, and 20% DMSO for 4 h at 37°C followed by a blocking step in PBS containing 0.2% Triton X-100, 10% DMSO, 6% horse serum and 0.05% sodium azide during 4 h at 37°C. Larvae were then stained for 3 days at 37°C with primary antibodies (chicken polyclonal

anti-GFP, A10262, Thermo Fisher Scientific, dilution 1:500; rabbit polyclonal anti-mCherry, 600-401-P16, Rockland Immunochemicals, dilution 1:500; and a mouse monoclonal anti-$N_{SVCV}$, Bio331-Moab a-SVC, BioX Diagnostics, dilution 1:50) diluted in PBS containing 0.2% Tween, 5% DMSO, 10 μg/mL heparin, 3% horse serum and 0.05% sodium azide. Following staining with primary antibodies, larvae were rinsed 3 times with PBS supplemented with 0.1% Tween for 15 min. Secondary antibodies AlexaFluor 488 goat anti-chicken (Thermo Fisher Scientific, A-11039), AlexaFluor 594 goat anti-rabbit (Thermo Fisher Scientific, A-11037), and AlexaFluor 647 goat anti-mouse (Thermo Fisher Scientific, A21240) were diluted in the same solution than primary antibodies and applied during 3 days at 37˚C. Larvae were rinsed 3 times in PBS containing 0.1% Tween for 15 min at room temperature (RT). Finally, immunos-tained larvae were cleared with a Refractive Index Matching Solution (RIMS; Histodenz, Sigma-Aldrich [57]) or 80% of glycerol.

## Zebrafish larvae imaging

Stereomicroscopy of anesthetized larvae was performed with a Leica MZ10F microscope equipped with an external light source (Leica EL6000), a PLAN APO 1× objective and a MC170HD camera (Leica microsystems).

For *in vivo* confocal imaging, anesthetized larvae were transferred to 3% agarose casts containing wells allowing to position larvae. Larvae were embedded in 0.5% low-gelling agarose (Sigma-Aldrich). The orientation of larvae was adjusted to lateral view before complete gelation. E3 medium supplemented with eugenol was added to embedded larvae. Confocal images of larvae were acquired with a Leica SP8 confocal/two-photon microscope using a HC PL FLUOTAR 10x/0.30 (#11506505, Leica) and an HCX IRAPO L 25×/0.95 water immersion objective (#11506340, Leica microsystems).

For confocal imaging of fixed samples, larvae were mounted in RIMS containing 0.8% of low melting agarose or 80% glycerol under coverslips #1 with homemade spacers.

## Laser induced injury

Anesthetized larvae were mounted as described above. The injury was performed with a fem-tosecond TI:Sapphire laser (Chameleon Vision II laser, Coherent) whose wavelength was adjusted to 800 nm. A crescent moon shape area was specified just above the urogenital aper-ture using the Navigator module of the LASX software which allows multi-tiles acquisition. The focal plane was adjusted to target the epidermis of the trunk. The targeted region was scanned with an 800 ns pixel dwell time. Laser intensity was set at 30%. The presence of a wound was evidenced by the formation of cavitation bubbles following laser illumination. Immediately after wounding, larvae were carefully removed from low-gelling agarose under a stereomicroscope and processed as described in the rSVCV-mCherry infection section. Larvae were then used for stereomicroscope acquisition or real-time confocal acquisitions. Two mock-infected and two rSVCV-mCherry larvae were mounted for confocal *in vivo* imaging as described above. Real-time acquisitions of neutrophil recruitment on the site of injury were performed overnight starting 2 hours post-infection. For each larva, a stack of images was acquired every 2 minutes during 18 h.

## Image analyses

Images were processed with Fiji (https://imagej.net/software/fiji/). Brightness and contrast of images were adjusted using the same parameters for each condition. For quantification of neu-trophil recruitment after laser photoablation, a ROI was delimited around the laser-induced

injury on the first frame of the time-lapse. Mean gray value of the GFP signal inside the ROI was measured for each time step. Data were normalized to percentages for plotting.

## Complex Object Parametric Analyzer and Sorter (COPAS)

The COPAS XL (Union Biometrica) large particle sorter was used for the analysis of 30–38 mock-infected or rSVCV-mCherry infected zebrafish larvae at 0, 6, 24, and 48 hours post infection. Analyses are based on size, optical density and fluorescence intensity. The COPAS FP 1000 configuration contains a flow cell, optics, and fluidic components designed to accommodate objects of 30–750 microns in diameter and is optimized for objects from 200 to 700 microns in diameter. It is equipped with 488 nm and 561 nm Solid State lasers and emission filter set of which the one for the detection of the mCherry signal was chosen. The acquisition parameters were: optical density threshold (extinction) of 230 mV (COPAS value: 20); minimum time of flight of 270 ms (COPAS value: 700); red photomultiplier tube (PMT) voltage of 380 V. The Profiler tool was used to analyze all data points per conditions for extinction and fluorescence intensity. This allows the detection of the distribution of mCherry signals, visualized as peaks within the fish body. Data were analyzed using the whole-body integrated value of mCherry fluorescence in fish larvae.

## Statistical analyses

The results of the individual red fluorescence of each larva detected using the COPAS system and the results of the gene expression analyzed by RT-qPCR were represented graphically as the mean ± standard error of the mean (SEM), and significant differences were obtained using Student's test and displayed as ***, p value < 0.001; **, p value < 0.01; and *, p value < 0.05. The correlation between the total red fluorescence signal detected in infected larvae by the COPAS system and the viral titer obtained by plaque assay in EPC cells was evaluated using Pearson's correlation coefficient and is expressed as $r^2$.

## Supporting information

**S1 Fig. Antigenomic SVCV infectious cDNA construct.** The details of the design of infectious SVCV cDNA are described in Materials and Methods. Shown are *XhoI* and *SmaI* sites that were introduced into the M-G, G-L intergenic regions, respectively and *EagI* site that was introduced in the trailer region: the sequence and nucleotide position of each site in the antigenome are indicated, with nucleotide substitutions made to create the sites in bold and the wild-type sequence shown above.
(TIF)

**S2 Fig. Insertion of additional expression cassettes in SVCV genome.** A. Alignment of SVCV intergenic regions. The complete intergenic regions between N-P, P-M, M-G and G-L genes of 16 SVCV strains were aligned to define the minimal consensus sequence DDDRTAT-GAAAAAAACTAACAGASATCATG (with D = G, A or T, R = G or A, and S = G or C). This untranslated region is composed of the transcriptional termination/polyadenylation gene end (GE) signal, TATGAAAAAAA, and the transcription initiation gene start (GS) signal, AACA-GASATCATG (with S = G or C). B. Details of the intergenic regions between M-G and G-L genes in the wild-type SVCV sequence. Stop codons are underlined and gene end signals are boxed. C. Details of the insertion of an additional gene in the M-G and G-L intergenic regions with the creation of a *XhoI* and *SmaI* unique restriction sites, respectively. Additional GE and GS signals are indicated in red upstream of the additional gene (gene of interest).
(TIF)

**S3 Fig. Genetic stability of mCherry reporter gene inserted in rSVCV genome.** rSVCV-mCherry was passaged up to 10 times on EPC cells. At the tenth passage, viral RNA was extracted from infected cell supernatants to confirm the presence of the mCherry expression cassette and its sequence by Sanger sequencing (A). RT-PCR products amplified with specific primers (5_SvcvCasMG; CATCAACATGGATACAACGGGATGG and 3_SvcvCasMG; CATTCAGCTGCATGGCAGATCCATC) from wild-type rSVCV (rSVCV wt) and rSVCV-mCherry (Passage 10) were analyzed on a 1% agarose gel. The sizes of the bands of the ladder (L) and the specific PCR fragments are indicated on the left and on the right, respectively. rSVCV-mCherry positive EPC cells at passage 10 (B). The cells were incubated at 25°C for 32 hours. Live cell monolayers were then visualized with a UV-light microscope. Scale bars, 100 μm.
(TIF)

**S4 Fig. Stereomicroscopic observations of rSVCV-mCherry in the caudal fin of bath-infected carp.** A. Visualization of rSVCV-mCherry at day 3 post infection by stereomicroscopy in the caudal fin of bath-infected carp. Infected foci (red) are located at the margin of the caudal fin and between bony rays (interray). Scale bar: 1 mm. B. Magnified view of inset depicted in A showing infection of interray. Scale bar: 200 μm.
(TIF)

**S5 Fig. Bioluminescence emission of rSVCV-ffLUC or rSVCV-akaLUC *in vivo* in carp after substrate injection intraperitoneally (IP).** A. Carp (mean weight, 1.91 g) were divided into two groups with one group that had their caudal fins cut (cut tail) prior to infection by immersion with rSVCV-ffLUC G/L. At 3 days post infection, 4 fish in each group were randomly harvested, anesthetized, and IP injected with 50 μL of luciferin (30 mg/mL) before imaging using an IVIS Spectrum BL imaging system. Stars indicate carp with detectable bioluminescent foci. B. Carp (mean weight, 1.91 g) were divided into two groups as described above prior to infection by immersion with rSVCV-akaLUC G/L. At 7- and 24- days post infection, 2 fish in each group were randomly harvested, anesthetized, and IP injected with 50 μL of Akalumine-HCl substrate (1 mM) before imaging. Stars indicate carp with detectable bioluminescent foci.
(TIF)

**S6 Fig. Comparison of two routes of substrate delivery for bioluminescence emission by rSVCV-akaLUC *in vivo* in carp at day 24 post infection.** A. Delivery of Akalumine-HCl substrate by IP injection (50 μL/fish of 1 mM Akalumine-HCl). The same fish before (both sides) and after dissection is shown in panel A. B. In order to compare both bioluminescence profiles, the picture of the fish shown in Fig 4 and corresponding to the delivery of Akalumine-HCl substrate by bath immersion for 2 h at 10°C (40 mM diluted at 1/10,000 in water) was included in this supplementary figure. The same fish before and after dissection is shown in panel B.
(TIF)

**S7 Fig. Juvenile zebrafish model of rSVCV-mCherry infection by bath immersion.** A. Nacre zebrafish juveniles (n = 51) at 25 dpf were infected by bath immersion with rSVCV-mCherry ($1 \times 10^7$ PFU) and incubated at 24°C. Mortality was recorded daily and is presented as the mean of the cumulative percent of dead larvae recorded in two independent experiments. No mortalities were recorded in the mock-infected group (n = 25). B. Virus load in individual juveniles. At different times post infection, juveniles (n = 10 in 2 independent infections) were randomly harvested and virus load was determined by plaque titration in EPC cells. Means are shown together with standard errors. C. Detail of mCherry fluorescent foci at 24 hpi in different infected juveniles. Pictures are presented for mock infected (top panel) and SVCV infected fish (bottom panel) in the following order: brightfield and red fluorescence.

Red arrows indicate the areas of infection. Scale bars: 500 μm.
(TIF)

**S8 Fig. rSVCV-mCherry infection by microinjection in zebrafish larvae.** A. Mortality is presented as the mean of cumulative percent of dead larvae recorded as well as the percent of mCherry positive larvae. No mortality was recorded in the mock-infected group. B. rSVCV-mCherry replication in zebrafish larvae microinjected in the duct of Cuvier. 5 groups of 5 larvae were randomly harvested at different times after infection (0, 6, and 24 hpi) and gene expression was analyzed by RT-qPCR. Virus loads are expressed as ratio of mRNA copy of SVCV nucleoprotein to ef1a housekeeping gene. Means are shown together with standard errors. C. Examples of zebrafish larvae microinjected in the duct of Cuvier at 24 hpi. Micrographs of mock- and rSVCV-mCherry infected larvae are presented in brightfield and red fluorescence. Scale bars: 200 μm.
(TIF)

**S9 Fig. Fluorescence analysis in rSVCV-Cherry infected larvae using the COPAS system.** Dot plots of the red fluorescence signal of mock- and rSVCV-mCherry-infected larvae recorded with the COPAS system at different times post infection (6, 24, and 48 hpi). The R3 region was drawn around the mock-infected larvae to define the background signal for mCherry (autofluorescence) and the R4 region was delineated in rSVCV-mCherry infected larvae to determine the positive signal for mCherry.
(TIF)

**S10 Fig. Recruitment of immune cells in SVCV-infected larvae.** A. Visualization of macrophages (green) at 24 hpi by confocal microscopy in the *Tg(mpeg1:eGFP)gl22* transgenic line infected with rSVCV-mCherry (red). Blue arrows show sites of viral replication. Scale bars: 200 μm. B. Area of the trunk of *Tg(mpeg1:eGFP)gl22* transgenic larvae in which mock-infected larvae macrophages show a typical ameboid shape and rSVCV-infected larvae macrophages have rounded shape even in an area located far from the virus replication sites. Scale bars: 100 μm. C. Visualization of neutrophils (green) at 24 hpi by confocal microscopy in the *Tg(mpx:GFP)[i114]* transgenic line infected with rSVCV-mCherry (red). Blue arrows show sites of viral replication. Scale bars: 200 μm.
(TIF)

**S11 Fig. Stereomicroscopic observations of tail lesions in rSVCV-infected larvae at 24 hpi.** Micrographs of mock- and rSVCV-infected larvae at 24 hpi taken with a stereomicroscope. Micrographs in brightfield with modified contrast to perceive the fin fold. Head arrows point to small wounds in the SVCV-infected larvae that were not observed in mock-infected larvae. Scale bars: 200 μm.
(TIF)

**S12 Fig. Neutrophil recruitment in the photoablation (PA) site at 24 h post SVCV infection.** Brightfield and fluorescence microscopy images (mpx:GFP, neutrophils in green; mCherry, rSVCV in red) of SVCV- and mock-infected Tg(mpx:GFP)[i114] larvae at 26 hpPA and 24 hpi. The photoablation area is marked with a yellow dotted crescent, and infected sites far from this area are pointed with yellow arrows. For rSVCV-infected fish #4–6, other primary sites of infection are shown. Scale bars: 200 μm.
(TIF)

**S1 Video. rSVCV-mCherry infection of the caudal fin of bath-infected carp.** Confocal imaging of the caudal fin of carp infected by bath with rSVCV-mCherry. rSVCV infected cells are visualized in red and nucleus stained with DAPI in blue. Z-stack showing infection of most superficial cells in caudal fin. Scale bar: 20 μm.
(AVI)

**S2 Video. IL1β expression in rSVCV-infected larva at 24 hpi.** Confocal imaging of the caudal fin of a transgenic Tg(il1β:GFP-F) larva infected with rSVCV at 24 hpi. IL1β expressing cells in green and rSVCV-mCherry infected cells in red.
(MP4)

**S3 Video. IL1β expression in rSVCV-infected larva at 48 hpi.** Confocal imaging of the caudal fin of a transgenic Tg(il1β:GFP-F) larvae infected with rSVCV at 48 hpi. IL1β expressing cells in green and rSVCV-mCherry infected cells in red.
(MP4)

**S4 Video. Neutrophil trafficking in the photoablation (PA) site at 24 h post rSVCV infection.** Time-lapse recording of the PA site of mock -infected transgenic Tg(mpx:GFP)[i114] larvae from 4 hpPA and 2 hpi to 22 hpPA and 20 hpi. Neutrophils (mpx:GFP) are in green and rSVCV-infected are in red (mCherry signal).
(MP4)

**S5 Video. Neutrophil trafficking in the photoablation (PA) site at 24 h post rSVCV infection.** Time-lapse recording of the PA site of mock-infected transgenic Tg(mpx:GFP)[i114] larvae from 4 hpPA and 2 hpi to 22 hpPA and 20 hpi. Neutrophils (mpx:GFP) are in green and rSVCV-infected are in red (mCherry signal).
(MP4)

**S6 Video. Neutrophil trafficking in the photoablation (PA) site at 24 h post rSVCV infection.** Time-lapse recording of the PA site of rSVCV-infected transgenic Tg(mpx:GFP)[i114] larvae from 4 hpPA and 2 hpi to 22 hpPA and 20 hpi. Neutrophils (mpx:GFP) are in green and rSVCV-infected are in red (mCherry signal).
(MP4)

**S7 Video. Neutrophil trafficking in the photoablation (PA) site at 24 h post rSVCV infection.** Time-lapse recording of the PA site of rSVCV-infected transgenic Tg(mpx:GFP)[i114] larvae from 4 hpPA and 2 hpi to 22 hpPA and 20 hpi. Neutrophils (mpx:GFP) are in green and rSVCV-infected are in red (mCherry signal).
(MP4)

**S1 Info. Entire raw data.**
(XLSX)

## Acknowledgments

We are grateful to members of the fish facilities for taking care of experimental fish (IERP-UE907, Jouy-en-Josas Research Center, France doi.org/10.15454/1.5572427140471238E12) which belongs to the National Distributed Research Infrastructure for the Control of Animal and Zoonotic Emerging Infectious Diseases through In Vivo Investigation (EMERG'IN DOI: doi.org/10.15454/90CK-Y371). We also thank the IERP-EMERG'IN platform for access to the IVIS Spectrum and COPAS, which were both financed by the région Ile-de-France (DIM-1Health).

We thank Louison Beugnies for her excellent technical support.

## Author Contributions

**Conceptualization:** Sandra Souto, Raquel Lama, Maxence Frétaud, Jean K. Millet, Christelle Langevin, Stéphane Biacchesi.

**Formal analysis:** Sandra Souto, Raquel Lama, Emilie Mérour, Manon Mehraz, Maxence Frétaud, Jean K. Millet, Christelle Langevin, Stéphane Biacchesi.

**Funding acquisition:** Christelle Langevin, Stéphane Biacchesi.

**Investigation:** Sandra Souto, Raquel Lama, Emilie Mérour, Manon Mehraz, Julie Bernard, Annie Lamoureux, Sarah Massaad, Maxence Frétaud, Dimitri Rigaudeau, Jean K. Millet, Christelle Langevin, Stéphane Biacchesi.

**Methodology:** Sandra Souto, Raquel Lama, Emilie Mérour, Manon Mehraz, Maxence Frétaud, Dimitri Rigaudeau, Jean K. Millet, Christelle Langevin, Stéphane Biacchesi.

**Supervision:** Christelle Langevin, Stéphane Biacchesi.

**Writing – original draft:** Christelle Langevin, Stéphane Biacchesi.

**Writing – review & editing:** Sandra Souto, Raquel Lama, Emilie Mérour, Manon Mehraz, Maxence Frétaud, Dimitri Rigaudeau, Jean K. Millet, Christelle Langevin, Stéphane Biacchesi.

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
