## [Decision Letter · Decision Letter 0]

18 Mar 2024

Dear Dr Biacchesi,

Thank you very much for submitting your manuscript "In vivo multiscale analyses of spring viremia of carp virus (SVCV) infection: from model organism to target species" for consideration at PLOS Pathogens. As with all papers reviewed by the journal, your manuscript was reviewed by members of the editorial board and by several independent reviewers. In light of the reviews (below this email), we would like to invite the resubmission of a significantly-revised version that takes into account the reviewers' comments.

The design and development of recombinant Spivivirus cyprinus/SVCV strains and the tracking of the bioluminescent virus in infection models (both zebrafish and natural host) in vivo leads to key biological observations, notably about the entry pathway of the virus. The model appears elegant as it provides a unique opportunity to visualize in real time the virus in infected cells and to analyze the subsequent immune response.

Nonetheless, authors should address the main points raised by reviewer 1, namely about the genetic stability of the recombinant strains that are produced and about the absence of virus detection in some tissues. Some additional controls are mandatory to answer these points. Answers to the minor points addressed by the two reviewers should also be provided.

We cannot make any decision about publication until we have seen the revised manuscript and your response to the reviewers' comments. Your revised manuscript is also likely to be sent to reviewers for further evaluation.

Sincerely,

Florence Margottin-Goguet

Academic Editor

PLOS Pathogens

Matthias Schnell

Section Editor

PLOS Pathogens

Michael Malim

Editor-in-Chief

PLOS Pathogens

orcid.org/0000-0002-7699-2064

Reviewer's Responses to Questions

**Part I - Summary**

Reviewer #1: In this manuscript Souto et al reported the first reverse genetics system for Sprivivirus cyprinus (previously called Spring viremia of carp virus (SVCV)), the use of this system for production of recombinant viruses expressing reporter genes, and the characterization of the infection by some of the recombinants using two in vivo infection models (common carp and zebrafish). This study led to key biological observations among which the most interesting is probably the identification of the skin epidermis as the major portal of entry of SVCV into its host. This study provides one more evidence that the skin epidermis of teleost fishes acts as a major portal of entry of viruses belonging to distant phylogenetic classes. This study represents a significant amount of data that will interest the scientific community working on fish infectious diseases and using zebrafish as a laboratory model. Few key experiments must be performed to support the conclusions of this MS. Other minor aspects of the MS should be addressed to further increase its quality.

Reviewer #2: Spring viremia of carp virus (SVCV) is an important fish virus that is a threat to aquaculture and ecosystems. The authors describe their work in developing recombinant SVCV strains that express fluorescent and luminescent proteins that enable tracking of viral infections. With these rSVCV strains, the authors demonstrate the arc of SVCV infection using immersion models in carp and immersion and intravenous injection models in zebrafish. The authors replicate aspects of natural infection, and thus the infection models they developed have great potential in understanding the course of SVCV infection and the immune response fish species mount to these infections. In addition, these infection models may help develop therapeutic strategies that may confine and/or eliminate SVCV infections. The work described in this manuscript is significant and has the potential to move the field forward. Overall, the general execution and scholarship is good, save for a few concerns about how the data are presented and the controls used in some experiments.

**Part II – Major Issues: Key Experiments Required for Acceptance**

Reviewer #1: 1/ A key point in this study that needs to be addressed is the genetic stability of the recombinant produced when passed in cell culture or in vivo. Indeed, the present study demonstrated that insertion of reporter sequence into the genome of the parental strain led to a reduction of viral fitness (Figure 3). Consequently, mutations leading to the loss of the reporter sequence would inevitably lead to the selection of such mutants in the population of virions. Of course, such mutants would be “invisible” as they do not express the reporter gene but they could affect the biological observations through “quasi-species” phenomena. It is also possible that transcomplementing mutations could be selected by passage in vitro and/or in vivo. The kinetics of mortality reported in Figure 3 for the recombinants supported these hypotheses. Of note, while the virulence of the rSVCVffluc recombinants does not seem to be affected by the insertions based on final mortality rates, it is astonishing that the appearance of the first dead fish occurred after 3 to 4 times longer incubation period compared to the parental control. This result could be explained by progressive selection of mutants according to time.

To address these hypotheses, the present study must address genomic (sequence) and phenotypic (reporter expression, viral replication in cell culture (growth, plaque size)) stability throughout passages both in cell culture and in vivo (fish infection).

2/ Another key point to keep in mind when using bioluminescent or fluorescent reporters is that “what you detect is real but it is possible that the presence of the virus is undetectable in some tissue for different reasons (access of substrate, expression of the reporter, access of excitation light in case of fluorescence …)”. When using bioluminescent reporters, two obvious points must be addressed. Does the substrate reach all tissues? Is there any structure preventing the light emitted by the tissues to reach the detector. In relation to these two questions, the present study will benefit from data resulting from injections of the luciferase substrates (both) and the use of the recombinants expressing the two different luciferases used. Also, ex vivo bioluminescent analyses directly exposing organs to the detector are mandatory before concluding that they are negative for viral replication. This comment is particularly important for gills.

3/ Concerning the demonstration that the skin acts as the major portal of entry of the virus into its host, it will be very interesting to demonstrate at the subcellular level that epidermal cells are indeed subject to viral infection. A former study on viral entry into fish adopted this approach by electronic microscopy taking profit of the bioluminescent signal to target sample collection. Similarly, observation in zebrafish should be further supported by data demonstrating at the histological level the infection of the epidermis.

Reviewer #2: There are a few figures that lack control data, including figure 5 and supplementary figure 3. It seems that it would be appropriate to show negative control data within these figures. While figure 2 shows that mock infection does not cause any mortality, mock infection data should be shown in figure 5 and supplementary figure 3.

**Part III – Minor Issues: Editorial and Data Presentation Modifications**

Reviewer #1: 1/ Spring viremia of car virus (SVCV) is the common name used to describe the virus subject of the present study. However, this name has been changed by the ICTV in 2015 to Sprivivirus cyprinus (https://ictv.global/taxonomy/taxondetails?taxnode_id=202201765&taxon_name=Sprivivirus%20cyprinus). It seems important to his reviewer to mention or even to use the official name of this virus in this MS. To provide the best visibility to this MS, it is suggested to mention both the former and current virus nomenclature in the MS but also in related links.

2/ Throughout the MS, there are repetitions that should be deleted. This is particularly the case of the discussion which repeats sections of the results.

3/ This MS will benefit from a systematic presentation of all the recombinant (structure and data). See for example figure 2A which presents only three of the recombinants produced.

4/ At the beginning of the results (Figure 1), the authors justify sequencing and phylogenetic analyses as the basis for the rational selection of the Fijan strain as the parental strain for construction of the recombinants (see line 139-142). “To select a representative genome sequence for establishing a robust reverse genetics system …”. Surprisingly, this strain appears distant from most strains which share a monophyletic origine. Consequently, this section sounds paradoxical to the reader. It must be modified.

5/ Line 141,171, the authors seem to quote inconsistent numbers of genome sequences, 16 versus 14. Please double check and correct if necessary. In case the numbers are correct, please provide a short explanation in the MS.

6/ Figure 7B. It will be interesting to present the data using 3D graphics so that correlations between expression could be established (e.g. are the fish positive at the head systematically positive for the tail …).

7/ Figure 9. Is it possible to provide the data for fish inoculated by IV and analysed at 48hpi?

Reviewer #2: Overall, the manuscript is written well, but there are several issues with grammar and syntax, including punctuation, missing words, and word choice. This reviewer respectfully asks the authors to better edit their manuscript. Perhaps they could consider working with a scientific copy editor.

The introduction is quite expansive and could be made more concise.

Additional comments follow:

Line 35: missing word in the sentence?

Line 57: missing word in the sentence?

It would be interesting to see the three nucleotide mutations annotated in figure 1B.

It would be interesting to see the sequence alignment of the 16 SVCV strains with features annotated, perhaps included in the supplementary information.

In figure 2B, it would be helpful to see all five pSVCV constructs rather than just the three that were shown.

Line 197-198: This sentence is confusing as to what was done. Can this be clarified? E.g., how long were they incubated after transfection?

Line 244: is it 2 out of 4 fish as 3 dpi, as it appears to be 3 out of 4 in the corresponding figure?

Line 244-245: “out to” could be replaced by “of.”

Line 258: “privileged” does not seem to be the correct word choice.

Line 315: Please include references for this infection method.

Line 349-350: Authors should consider applying the appropriate gene names for il1b and tnfa

Lines 352-355: Was the expression of other zebrafish interferons considered? See Aggad et al or Varela et al.

Line 367: Mention transparency here. Other places, translucency is mentioned. Authors should pick one and be consistent.

Line 438: “impedes” may not be the best word choice.

Suppl. Fig 6B: Is there a way to quantify the shift in macrophage morphology? It could be an even more powerful way of representing these data.

Line 588-589: Authors show reduction of mpeg1 expression, not macrophage depletion directly. Do they have a direct method to test depletion (e.g., FACS)?

The y axis scaling for many of the panels in figure 9 is distorted. It appears that panels in figure 9A, 9C, and 9D can be rescaled for 20 to <5.

Line 542. Author should consider writing out full gene names for nfkb2 and relb rather than the shorthand they appear to use.

Line 599: Typo? “te”

Line 607: “impedes” may not be the best word choice.

Line 614: Spelling of “correlated.”

Reviewer #1: No

Reviewer #2: No
---

## [Decision Letter · Decision Letter 1]

7 Jun 2024

Dear Dr Biacchesi,

We are pleased to inform you that your manuscript 'In vivo multiscale analyses of spring viremia of carp virus (SVCV) infection: from model organism to target species' has been provisionally accepted for publication in PLOS Pathogens.

Best regards,

Florence Margottin-Goguet

Academic Editor

PLOS Pathogens

Matthias Schnell

Section Editor

PLOS Pathogens

Michael Malim

Editor-in-Chief

PLOS Pathogens

orcid.org/0000-0002-7699-2064

Reviewer Comments (if any, and for reference):

Reviewer's Responses to Questions

**Part I - Summary**

Reviewer #1: These elements were described earlier. This is a very interesting study related to host-pathogen interactions.

**Part II – Major Issues: Key Experiments Required for Acceptance**

Reviewer #1: The authors performed the experiments suggested.

**Part III – Minor Issues: Editorial and Data Presentation Modifications**

Reviewer #1: None

PLOS authors have the option to publish the peer review history of their article (what does this mean?). If published, this will include your full peer review and any attached files.

Reviewer #1: No

---

## [Editor Report · Acceptance letter]

30 Jul 2024

Dear Dr. Biacchesi,

We are delighted to inform you that your manuscript, "In vivo multiscale analyses of spring viremia of carp virus (SVCV) infection: from model organism to target species," has been formally accepted for publication in PLOS Pathogens.

Best regards,

Michael Malim

Editor-in-Chief

PLOS Pathogens

orcid.org/0000-0002-7699-2064